# Oxytocin modulates human chemosensory decoding of sex in a dose-dependent manner

Kepu Chen[1†], Yuting Ye[2†], Nikolaus F Troje[3], Wen Zhou[1,2,4*]

[1]State Key Laboratory of Brain and Cognitive Science, CAS Center for Excellence in Brain Science and Intelligence Technology, Institute of Psychology, Chinese Academy of Sciences, Beijing, China; [2]Department of Psychology, University of Chinese Academy of Sciences, Beijing, China; [3]Centre for Vision Research, York University, Toronto, Canada; [4]Chinese Institute for Brain Research, Beijing, China

**Abstract** There has been accumulating evidence of human social chemo-signaling, but the underlying mechanisms remain poorly understood. Considering the evolutionarily conserved roles of oxytocin and vasopressin in reproductive and social behaviors, we examined whether the two neuropeptides are involved in the subconscious processing of androsta-4,16,-dien-3-one and estra-1,3,5 (10),16-tetraen-3-ol, two human chemosignals that convey masculinity and femininity to the targeted recipients, respectively. Psychophysical data collected from 216 heterosexual and homosexual men across five experiments totaling 1056 testing sessions consistently showed that such chemosensory communications of masculinity and femininity were blocked by a competitive antagonist of both oxytocin and vasopressin receptors called atosiban, administered nasally. On the other hand, intranasal oxytocin, but not vasopressin, modulated the decoding of androstadienone and estratetraenol in manners that were dose-dependent, nonmonotonic, and contingent upon the recipients' social proficiency. Taken together, these findings establish a causal link between neuroendocrine factors and subconscious chemosensory communications of sex-specific information in humans.

***For correspondence:**
zhouw@psych.ac.cn

[†]These authors contributed equally to this work

**Competing interests:** The authors declare that no competing interests exist.

## Introduction

The term 'pheromone' is derived from ancient Greek pherein 'to transfer' and hormōn 'to excite'. Pheromones are known as chemical signals that convey information between members of the same species (*Karlson and Luscher, 1959*). Less appreciated are their intricate interplays with the central neural and neuroendocrine systems. In insects and rodents, specific neuropeptides and hormones are critically involved in the production, release, as well as perception of pheromones (*Cusson and McNeil, 1989*; *Lin et al., 2016*; *Mugford and Nowell, 1971*; *Raina et al., 1989*; *Tang et al., 1989*). Little is known as to whether this link holds for non-rodent mammals or how it is manifested there. In humans, particularly, the very existence of pheromones has long been an issue of debate (*Wyatt, 2015*). Since the first report of human menstrual synchrony in 1971 (*McClintock, 1971*), nearly five decades of research has shown that human body secretions, particularly axillary sweat, subconsciously convey emotion and reproductive state (*de Groot et al., 2015*; *Stern and McClintock, 1998*; *Zhou and Chen, 2009*), which in turn are driven by complex neurochemical changes (*Blouin et al., 2013*; *Jones and Lopez, 2013*). Of the many components of human body secretions, two endogenous steroids, androsta-4,16,-dien-3-one, a non-androgenic derivative of gonadal progesterone (*Gower and Ruparelia, 1993*), and estra-1,3,5 (10),16-tetraen-3-ol, related to the estrogen sex hormones but with no known estrogenic effects (*Thysen et al., 1968*), emerge as candidates of human sex pheromones. Aside from affecting autonomic responses and mood states

in women and men, respectively (*Bensafi et al., 2004*; *Jacob and McClintock, 2000*; *Lundström et al., 2003a*; *Olsson et al., 2006*), androstadienone and estratetraenol are found to effectively convey sex-specific information and activate the hypothalamus, a structure critically involved in sexual reproduction (*Simerly, 2002*), in distinct patterns. Specifically, androstadienone signals masculinity and estratetraenol signals femininity to their targeted recipients (*Zhou et al., 2014*). They also prime the identification of emotionally receptive states for the potential mates with whom they are associated (*Ye et al., 2019*). In parallel, androstadienone activates the hypothalamus in heterosexual women and homosexual men, but not in heterosexual men or homosexual women, whereas estratetraenol activates the hypothalamus in heterosexual men and homosexual women, but not in heterosexual women or homosexual men (*Berglund et al., 2006*; *Savic et al., 2001*; *Savic et al., 2005*).

The hypothalamus contains several sexually dimorphic nuclei and is well documented to coordinate neuroendocrine responses with sensory cues that regulate motivated behavior (*Simerly, 2002*). It produces, among various hormones, two structurally similar nonapeptides called oxytocin and vasopressin that function both as neuropeptides and hormones and have ancient roles in reproductive behaviors. Oxytocin and vasopressin are heavily implicated in sexual arousal and subsequent copulatory behavior in rats, rabbits, rams, bulls, and humans, with their homologs mediating comparable activities in nematodes, snails, and bony fish (*Carter, 1992*; *Donaldson and Young, 2008*). Moreover, the two peptides, especially oxytocin, are believed to underlie the evolution and expression of human sociality including trust, love, and cooperation (*Carter, 2014*; *Donaldson and Young, 2008*). Based on the animal literature, receptors for oxytocin and vasopressin are expressed in the olfactory system and other downstream sexually dimorphic nuclei under the modulatory influence of gonadal steroids, and are involved in the processing of social odors (*Stoop, 2012*). These findings have led us to suspect that the effects of androstadienone and estratetraenol, considered by some as putative human sex pheromones, could be regulated by oxytocin and/or vasopressin.

To examine the above hypothesis, we employed a gender identification task of point-light walkers (PLWs) (*Figure 1A*)—dynamic point-light displays portraying the gaits of walkers—that had been shown to be sensitive to the interactive effects of androstadienone, estratetraenol, and the recipients' sex and sexual orientation (*Zhou et al., 2014*), and combined it with pharmacological manipulations of central oxytocin and vasopressin. We also measured each participant's social proficiency (the opposite of autistic-like tendency) with the Autism Spectrum Quotient (AQ) (*Baron-Cohen et al., 2001*), in view of the positive association between social proficiency and endogenous oxytocin level (*Koven and Max, 2014*; *Lancaster et al., 2015*; *Parker et al., 2014*) and research showing that individual differences in social proficiency or AQ score can moderate the effects of exogenously administered oxytocin (*Bartz et al., 2019*; *Bartz et al., 2010*; *Bartz et al., 2011*). From the data we obtained psychometric curves that depicted the probability of making male judgments as a function of the physical gender of the PLWs (*Figure 1B*) under different combinations of olfactory stimuli and drug treatments. Systematic comparisons of these psychometric curves (*Figure 1C*) across five experiments totaling 1056 testing sessions enabled us to assess the roles of oxytocin and vasopressin in chemosensory communications of masculinity and femininity through androstadienone and estratetraenol in both heterosexual (Experiment 1, sample size n = 72) and homosexual (Experiment 2, n = 72) men (*Figure 1D*) as well as in socially less proficient (Experiments 3 and 4, n = 24 in each experiment) and socially proficient (Experiment 5, n = 24) individuals (*Figure 1E*).

## Results

General procedures of the gender identification task have been described elsewhere (*Zhou et al., 2014*). In brief, genders of the PLWs were quantified as Z scores (*Troje, 2002*) and ranged in seven equal steps from feminine (−0.45 SD) to masculine (0.45 SD) with 0 marking the approximate gender-neutral point individually adjusted for each participant prior to the actual experiment in the absence of olfactory stimulus and drug treatment. In each trial of the task, participants viewed a PLW for 500 ms (0.5 walking cycle) and made a forced choice judgment on whether it was a male or a female walker (*Figure 1A*). To characterize how gender perception criteria were influenced by androstadienone or estratetraenol under different manipulations of oxytocin and vasopressin, we fitted gender judgments for all PLWs separately for each olfactory condition and each participant under each drug treatment condition with a Boltzmann sigmoid function that contained two

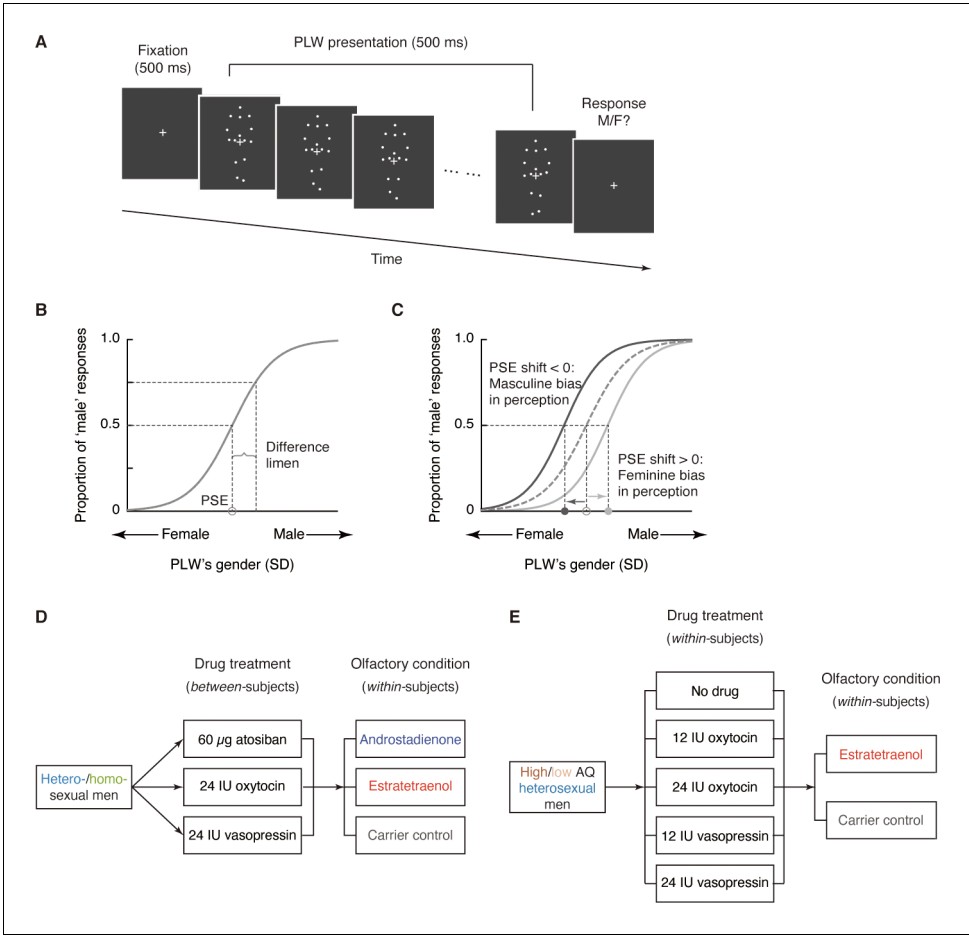

**Figure 1.** Experimental procedure and design. (**A**) Schematic illustration of a trial in the gender identification task. Each trial began with a 500 ms fixation cross, followed by a dynamic point-light walker (PLW) presented for 500 ms (0.5 walking cycle). Participants then pressed one of two buttons to indicate whether it was a male or a female walker. The physical gender of the PLW was denoted by a Z score and ranged in seven equal steps from feminine (−0.45 SD) to masculine (0.45 SD) with the center (0) individually adjusted to approximately perceived gender neutrality in the absence of drug treatment and olfactory stimulus. (**B**) Participants' gender judgments for PLWs were fitted with a psychometric function that contained two parameters: point of subjective equality (PSE) and difference limen. The PSE is the physical gender of a PLW (Z value) that corresponds to a probability of 50% on the fitted psychometric function (gray circle on the x-axis), where the participant perceived the PLW as equally masculine and feminine. The difference limen is half the interquartile range of the fitted function. (**C**) Participants' PSEs when smelling either chemosignal were compared with those when smelling the carrier control alone (gray circle). A negative PSE shift relative to the carrier control condition (dark gray arrow) corresponds to an overall leftward shift of the psychometric curve and reflects an increased tendency to judge the PLWs as male, hence a masculine bias in gender perception. Conversely, a positive PSE shift (light gray arrow) indicates a feminine bias in gender perception. All curves here are hypothetical. (**D**) Experiments 1 and 2 tested heterosexual and homosexual men, respectively. Drug treatment (60 μg atosiban, 24 IU oxytocin, 24 IU vasopressin) served as a between-subjects factor whereas olfactory condition (androstadienone, estratetraenol, carrier control) served as a within-subjects factor. Participants performed the experimental blocks of the gender identification task 35 min after nasal drug administration, under the continuous exposure of an olfactory stimulus. (**E**) Experiments 3 and 5 tested high AQ (AQ scores ≥ 25) and low AQ (AQ scores < 25) heterosexual men, respectively. Both drug treatment (no drug, 12 IU oxytocin, 24 IU oxytocin, 12 IU vasopressin, 24 IU vasopressin) and olfactory condition (estratetraenol, carrier control) were manipulated in a within-subjects fashion.

parameters: point of subjective equality (PSE) and difference limen (*Figure 1B*). The PSE, an index of response criterion, is the Z score (which denoted PLWs' physical gender) corresponding to a probability of 50% on the fitted psychometric function, where the participant perceived a PLW as equally masculine and feminine. The difference limen is half the interquartile range of the fitted function and

indicates response sensitivity. For each drug treatment condition, participants' PSEs when smelling androstadienone or estratetraenol—each dissolved in a clove oil carrier solution (1% v/v clove oil in propylene glycol)—were compared with those when smelling the clove oil carrier solution alone (control). A negative PSE shift relative to the carrier control condition corresponds to an overall left-ward shift of the psychometric curve and reflects an increased tendency to judge the PLWs as male, hence a masculine bias in gender perception (*Figure 1C*). Conversely, a positive PSE shift indicates a feminine bias in gender perception.

## Oxytocin, vasopressin, and subconscious chemosensory decoding of sex in heterosexual and homosexual men

We first recruited 72 heterosexual males (Kinsey scores = 0) in Experiment 1 and randomly assigned them to three groups of 24 each to receive different intranasal drug treatments: 24 IU oxytocin, 24 IU vasopressin, or 60 µg atosiban—a desamino-oxytocin analogue and a competitive antagonist of both oxytocin and vasopressin receptors (*Manning et al., 2012*). Nasally delivered oxytocin and vasopressin can directly access the cerebrospinal fluid (*Born et al., 2002*; *Freeman et al., 2016*); there has been some limited indication that intranasal atosiban is also centrally available (*Liu et al., 2018*; *Lundin et al., 1986*). In a double-blind procedure, each participant was tested in 3 sessions held at around the same time of the day on 3 consecutive days. During each session they performed the gender identification task (*Figure 1A*) while being continuously exposed to either androstadie-none (500 µM, 5 ml), estratetraenol (500 µM, 5 ml) or their carrier solution alone (control condition, 1% v/v clove oil in propylene glycol, 5 ml), one per session in a counterbalanced manner, 35 min fol-lowing drug administration (*Figure 1D*). The three olfactory stimuli were perceptually indiscriminable as assessed in a separate panel of 48 male participants (mean triangular discrimination accu-racy = 0.33 vs. chance = 0.33, $t_{47}$ = 0.02, p>0.9), which was also in line with earlier reports (*Ye et al., 2019*; *Zhou et al., 2014*). Our previous study had shown that heterosexual males were affected by estratetraenol, but not androstadienone, in making gender judgments. Specifically, estratetraenol subconsciously biased them toward perceiving the PLWs as more feminine (*Figure 2A*; *Zhou et al., 2014*). We reasoned that the effect of estratetraenol, if regulated by the oxytocin/vasopressin sys-tem, would be blocked by intranasal atosiban and modulated by intranasal oxytocin and/or vaso-pressin. It was difficult to predict the directions of their effects (if any), as both positive and negative effects of oxytocin and vasopressin have been reported in the literature, depending on dose, con-text, and personal characteristics (*Bartz et al., 2011*; *Carter, 2014*; *Donaldson and Young, 2008*). It was also possible that only one of oxytocin and vasopressin plays a role, in which case the chemo-sensory effect would not be altered by the administration of the other.

Indeed, in the atosiban-treated heterosexual men, we found that smelling estratetraenol relative to the carrier solution alone failed to influence their gender perception criteria (indexed by the PSEs, $t_{23}$ = 1.37, p=0.18; *Figure 2B,E*). Interestingly, this was also the case for those treated with 24 IU oxytocin ($t_{23}$ = 0.17, p=0.87; *Figure 2C,E*). Only in the participants treated with 24 IU vasopressin did estratetraenol induce a systematic bias towards perceiving the PLWs as more feminine ($t_{23}$ = 4.25, p<0.001, Cohen's d = 0.87; *Figure 2D,E*). The size of the estratetraenol-induced gender perception bias under vasopressin was similar to that found earlier without drug treatment (*Zhou et al., 2014*), and a direct comparison of the two sets of data showed no significant difference (p=0.35). In other words, vasopressin did not significantly enhance or diminish the effect of estrate-traenol on heterosexual males. On the other hand, smelling androstadienone relative to the carrier solution alone did not influence gender perception criteria regardless of drug treatments (i.e. 60 µg atosiban, 24 IU oxytocin, or 24 IU vasopressin, ps > 0.34; *Figure 2B–E*), consistent with earlier find-ings (*Zhou et al., 2014*).

These results presented a complex picture. Antagonizing oxytocin and vasopressin receptors with atosiban abolished the known effect of estratetraenol on heterosexual males (i.e. biasing them toward perceiving the PLWs as more feminine), yet 24 IU oxytocin appeared to produce the same effect. On the flip side, the administration of 24 IU vasopressin did not significantly alter the process-ing of estratetraenol in heterosexual males—they remained biased toward perceiving the PLWs as more feminine under the exposure of estratetraenol, to the same extent as when no drug was administered. We wondered if this pattern of drug influences would hold for the chemosensory decoding of masculine information carried by androstadienone. To this end, we turned to

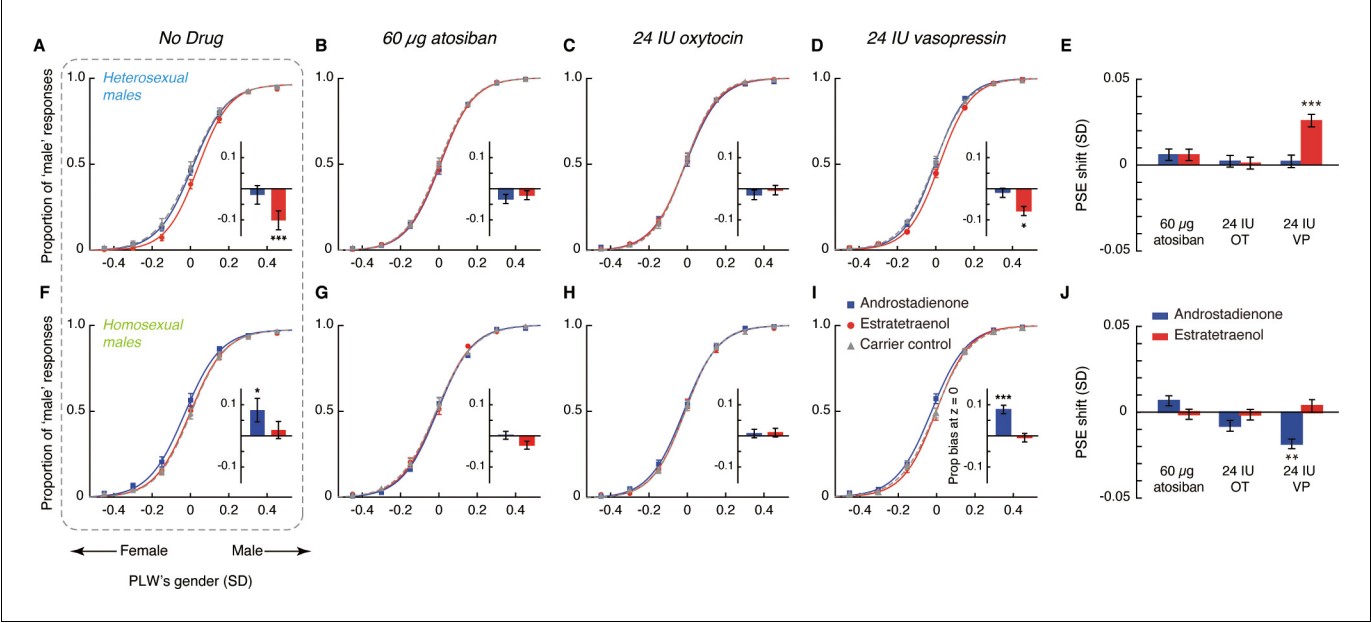

**Figure 2.** Oxytocin, vasopressin, and subconscious chemosensory decoding of sex in heterosexual and homosexual men. (**A–D, F–I**) Androstadienone- and estratetraenol- induced visual gender judgment biases in the absence of drug treatment (A and F, adapted from *Zhou et al., 2014* for comparison) and after the nasal administrations of 60 µg atosiban (**B, G**), 24 IU oxytocin (**C, H**), and 24 IU vasopressin (**D, I**) in heterosexual (Experiment 1, **A–D**) and homosexual (Experiment 2, **F–I**) men. Gender identification performances under the exposures of androstadienone, estratetraenol, and the carrier control are respectively fitted with sigmoidal curves (blue solid curves, red solid curves, and gray dashed curves, respectively). Insets show the androstadienone- and estratetraenol- induced proportional 'male' biases at the gender-neutral point of the point-light walkers (PLWs), that is, androstadienone- and estratetraenol- induced differences in the proportion of 'male' responses at Z = 0 relative to the carrier control condition. (**E, J**) Androstadienone- and estratetraenol- induced overall point of subjective equality (PSE) shifts with respect to the carrier control after the nasal administrations of 60 µg atosiban, 24 IU oxytocin (OT), and 24 IU vasopressin (VP) in heterosexual (**E**) and homosexual (**J**) men. A positive PSE shift indicates a feminine bias, that is, a bias toward perceiving the PLWs as more feminine, whereas a negative PSE shift indicates a masculine bias, that is, a bias toward perceiving the PLWs as more masculine. Dashed box: data from our earlier study (*Zhou et al., 2014*); error bars: SEMs adjusted for individual differences; *: p<0.05; **: p≤0.01; ***: p≤0.005.

The online version of this article includes the following source data for figure 2:

**Source data 1.** Experiments 1 and 2.

homosexual males in Experiment 2, who had been shown to be subconsciously biased by androstadienone, but not estratetraenol, in making gender judgments (*Figure 2F*; *Zhou et al., 2014*).

Except for the participants' sexual orientation (homosexual males, mean Kinsey score ± SD = 5.26 ± 0.63), Experiment 2 was identical to Experiment 1, and revealed a similar pattern of drug effects. Gender perception criteria were unaffected by the exposure to androstadienone, relative to the carrier control, in the homosexual men treated with 60 µg atosiban ($t_{23}$ = 0.88, p=0.39; *Figure 2G,J*) as well as in those treated with 24 IU oxytocin ($t_{23}$ = −1.29, p=0.21; *Figure 2H,J*). By contrast, in the 24 IU vasopressin group, smelling androstadienone induced a systematic bias toward perceiving the PLWs as more masculine ($t_{23}$ = −2.99, p=0.007, Cohen's d = 0.61; *Figure 2I,J*). The strength of the androstadienone-induced masculine bias under vasopressin was again not different from that found earlier (*Zhou et al., 2014*) in homosexual males receiving no drug treatment (p=0.18). Meanwhile, in line with earlier findings (*Zhou et al., 2014*), smelling estratetraenol did not influence gender judgements in these homosexual men irrespective of drug manipulations (ps > 0.58; *Figure 2G–J*).

An omnibus ANOVA of the PSEs from both Experiments 1 and 2 (olfactory condition × drug treatment × sexual orientation) identified a significant interaction between olfactory condition and drug treatment ($F_{4,\ 276}$ = 3.58, p=0.007, partial $\eta^2$ = 0.05), which reinforced that the effects of androstadienone and estratetraenol on gender judgments were modulated by the drug manipulations, and that this modulation was similar in both heterosexual and homosexual males (olfactory condition × drug treatment × sexual orientation: $F_{4,\ 276}$ = 0.80, p=0.53). To facilitate comparison,

we highlighted the central tendencies of the androstadienone- and estratetraenol-induced PSE shifts (x- and y- axes, respectively) under different drug treatments in *Figure 3* (see also *Figure 3—figure supplement 1*), generated by using a standard bootstrapping procedure (*Davison and Hinkley, 1997*; see Materials and methods). Under 60 µg atosiban and 24 IU oxytocin, neither androstadienone nor estratetraenol induced any significant change in gender perception criterion in heterosexual (cyan dots) or homosexual males (lime dots), and the bootstrapped sample means of the two groups of men overlapped around the origin. Conversely, under 24 IU vasopressin, they formed two discrete clusters: estratetraenol biased heterosexual, but not homosexual, males toward perceiving the PLWs as more feminine (cyan dots fell around the vertical axis on the positive side), whereas androstadienone biased homosexual, but not heterosexual, males toward perceiving the PLWs as more masculine (lime dots fell around the horizontal axis on the negative side). In addition, the difference limens of the heterosexual and homosexual males did not differ ($F_{1, 138} = 0.47$, $p=0.50$) and were comparable across olfactory conditions and drug treatments (olfactory condition × drug treatment: $F_{4, 276} = 1.19$, $p=0.31$; olfactory condition: $F_{2, 276} = 0.48$, $p=0.62$; drug treatment: $F_{2, 138} = 0.24$, $p=0.79$). So were their self-reported mood states on the Profile of Mood States (*McNair et al., 1971*) (POMS; total mood disturbance: ps = 0.84, 0.51 and 0.90, respectively; all subscales: ps > 0.05, corrected). Thus, it was the criterion (reflected in the PSEs) rather than the sensitivity of gender judgment (reflected in the difference limens) or transient mood state that was swayed by the interplays between the chemosignals and the drug manipulations.

The combined results of Experiments 1 and 2 thus suggested that the decoding of feminine information carried by estratetraenol and that of masculine information carried by androstadienone, while contingent upon the recipients' sexual orientation, were subserved by similar neuroendocrine mechanisms that were disrupted by intranasal atosiban—the competitive antagonist of both oxytocin and vasopressin receptors, as well as by 24 IU oxytocin, and were unaffected by 24 IU vasopressin. Since vasopressin seemed to exert no effect (participants' response patterns to the chemosignals under 24 IU vasopressin were comparable to those previously obtained without drug treatment), by deduction, such mechanisms involved oxytocin. The question remained as to why the administration of 24

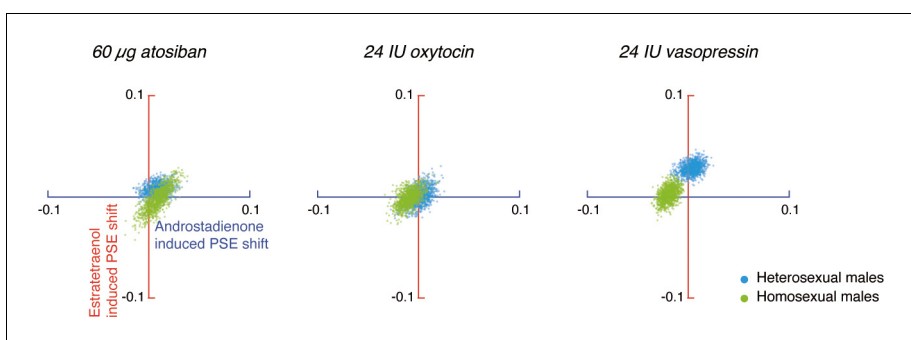

**Figure 3.** Central tendencies of androstadienone- and estratetraenol- induced point of subjective equality (PSE) shifts in heterosexual and homosexual men across drug conditions. Each subfigure shows the bivariate distributions of bootstrapped sample means for heterosexual men (1000 cyan dots) and homosexual men (1000 lime dots) plotted against the horizontal and vertical axes representing androstadienone- and estratetraenol-induced PSE shifts, respectively, following the nasal administration of 60 µg atosiban, 24 IU oxytocin, or 24 IU vasopressin. A positive value on either axis indicates a feminine bias, that is, a bias toward perceiving the point-light walkers (PLWs) as more feminine, whereas a negative value indicates a masculine bias, that is, a bias toward perceiving the PLWs as more masculine.

The online version of this article includes the following source data and figure supplement(s) for figure 3:

**Source data 1.** Experiments 1 and 2.

**Figure supplement 1.** Histogram distributions (with normal curves) of 1000 bootstrapped sample means for the points of subjective equality (PSEs) under each combination of olfactory and drug conditions in heterosexual and homosexual men.

**Figure supplement 2.** Central tendencies of the degrees of gender perception biases induced by chemosensory sexual cues in high AQ and low AQ individuals in Experiments 1 and 2.

IU oxytocin, like atosiban, exempted the participants from the influences of the chemosignals, and we explored it in more detail.

If, regardless of one's sexual orientation, oxytocin plays a role in the processing of chemosensory sexual cues associated with the preferred sex, given the relationship between social proficiency and endogenous oxytocin level (*Koven and Max, 2014*; *Lancaster et al., 2015*; *Parker et al., 2014*) and the heterogenous effects of intranasal oxytocin on individuals with different levels of social proficiency (*Bartz et al., 2011*), specifically as assessed by the AQ (*Bartz et al., 2019*; *Bartz et al., 2010*), it follows that individuals with different AQ scores could differ in their susceptibility to such chemosignals as well as to the effect of exogenous oxytocin. Put differently, social proficiency could be related to the subconscious extraction of chemosensory sexual information, a link hitherto unsuspected. As an initial attempt to test this inference, we reexamined the data from Experiments 1 and 2 to see if high AQ and low AQ participants differed in their susceptibility to the chemosignals. The participants' AQ scores ranged from 9 to 35. We adopted a relatively strict criterion—AQ scores $\geq$ 25, that is, 1 SD or more above the reported mean for males (*Baron-Cohen et al., 2001*)—for high AQ individuals in hopes to better capture the effects of social proficiency (*Bartz et al., 2019*). Whereas an AQ score above 32 is used as the cutoff for distinguishing individuals with clinically significant levels of autistic traits (*Baron-Cohen et al., 2001*), we did not opt for this stringent cutoff value here due to pragmatic considerations: Such individuals are rare in the general population and only 1 out of the 144 participants scored above 32. Those with AQ scores below 25 were classified as low AQ individuals. Overall, AQ scores were comparable among the drug groups and also between the heterosexual and the homosexual participants in Experiments 1 and 2 (drug treatment: $F_{2, 138} = 0.48$, p=0.62; sexual orientation: $F_{1, 138} = 1.39$, p=0.24; interaction: $F_{2, 138} = 0.097$, p=0.91). Since no significant effect of androstadienone or estratetraenol was observed under the 60 µg atosiban condition or 24 IU oxytocin condition, we focused our examination on the 24 IU vasopressin condition where the effects of the two chemosignals were comparable to those obtained earlier without drug treatment (*Zhou et al., 2014*). The majority (85.4%) of the participants were low AQ individuals (AQ scores < 25). Our supplementary analysis revealed that, relative to the high AQ participants (n = 7, 14.6%), estratetraenol and androstadienone induced larger gender perception biases in the low AQ participants, driving the overall effects of the chemosignals on gender perception (*Figure 3—figure supplement 2*). The results thus lent preliminary support to the postulated link between social proficiency and the subconscious processing of chemosignals, and led us to assess the effect of exogenous oxytocin separately in high AQ and low AQ individuals.

## Dose-dependent modulation of chemosensory decoding of sex by oxytocin but not vasopressin in high and low AQ individuals

It has been reported that the effect of intranasal oxytocin is more pronounced in socially less proficient individuals as measured by the AQ (*Bartz et al., 2019*; *Bartz et al., 2010*; *Bartz et al., 2011*) and that a lower dose of oxytocin could exert a more positive effect than a higher dose (*Cardoso et al., 2013*; *Quintana et al., 2017*; *Spengler et al., 2017*). To better probe the role of oxytocin in the chemosensory processing of sexual information and not miss any possible effect of vasopressin, we recruited 24 heterosexual males with high AQ scores (range: 25–36) in Experiment 3 and assessed the extent to which their gender perception was biased by estratetraenol under different doses of oxytocin and vasopressin in a within-subject design (*Figure 1E*). Specifically, each participant was tested in 10 sessions held around the same time of the day on 10 days that comprised of five drug conditions (2 sessions each): 12 IU oxytocin, 24 IU oxytocin, 12 IU vasopressin, 24 IU vasopressin, and no drug treatment. Under each drug condition, they performed the gender identification task while being continuously exposed to estratetraenol in one session, and to the carrier solution alone in the other session. Androstadienone was not included as it consistently showed no effect on heterosexual men's gender perception (Experiment 1) (*Zhou et al., 2014*).

In the absence of drug treatment, the high AQ heterosexual men failed to utilize the feminine information carried by estratetraenol in making gender judgments ($t_{23} = 0.70$, p=0.49; *Figure 4A,F*), which echoed with the preliminary result from the high AQ individuals in Experiments 1 and 2 (*Figure 3—figure supplement 2*). Remarkably, this chemosensory ability was restored by the administration of 12 IU ($t_{23} = 2.72$, p=0.012, Cohen's d = 0.55; *Figure 4B,F*), but not 24 IU ($t_{23} = -0.36$, p=0.72; *Figure 4C,F*), intranasal oxytocin. Meanwhile, the participants remained insusceptible to estratetraenol regardless of whether they were treated with 12 IU ($t_{23} = 0.76$, p=0.46; *Figure 4D,F*)

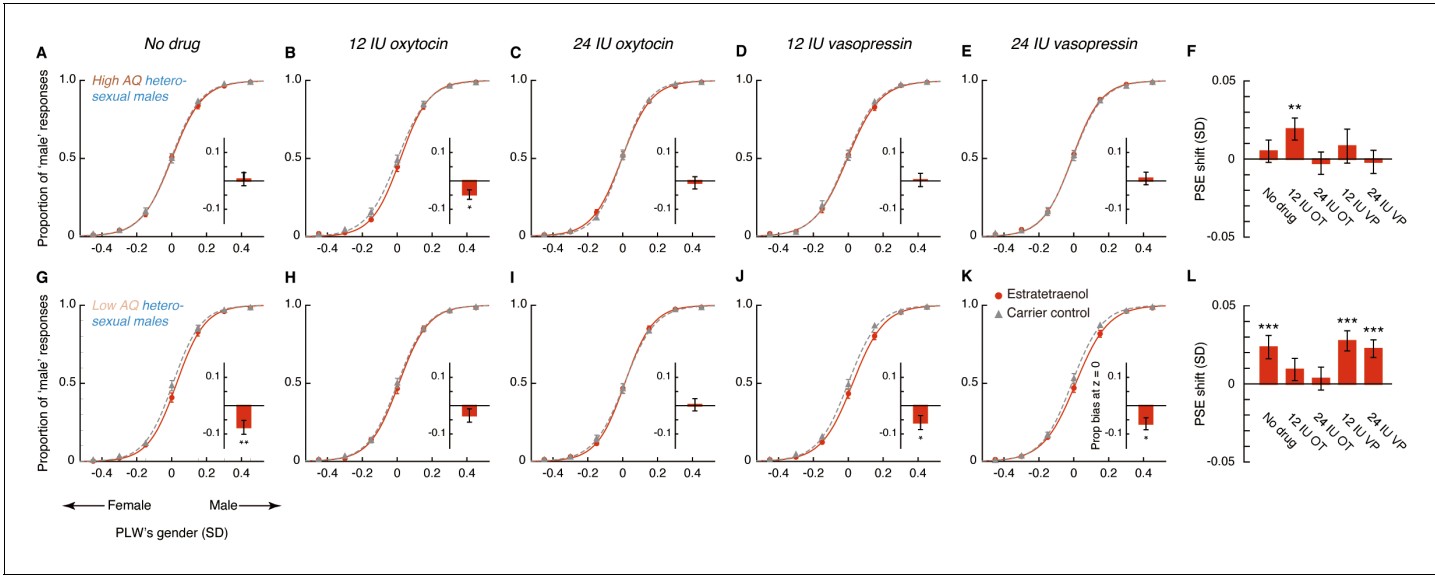

**Figure 4.** Oxytocin, but not vasopressin, modulates chemosensory decoding of femininity in heterosexual men in a dose-dependent manner. (A–E, G–K) Estratetraenol-induced visual gender judgment biases in the absence of drug treatment (A, G) and after the nasal administrations of 12 IU oxytocin (B, H), 24 IU oxytocin (C, I), 12 IU vasopressin (D, J), and 24 IU vasopressin (E, K) in high AQ (Experiment 3, A–E) and low AQ (Experiment 5, G–K) heterosexual men. Gender identification performances under the exposures of estratetraenol and the carrier control are fitted with sigmoidal curves (red solid curves and gray dashed curves, respectively). Insets show the estratetraenol-induced proportional 'male' biases at the gender-neutral point of the point-light walkers (PLWs), that is, estratetraenol-induced differences in the proportion of 'male' responses at Z = 0 relative to the carrier control condition. (F, L) Estratetraenol-induced overall point of subjective equality (PSE) shifts with respect to the carrier control in the absence of drug treatment and after the nasal administrations of 12 IU oxytocin, 24 IU oxytocin, 12 IU vasopressin, and 24 IU vasopressin in high AQ (F) and low AQ (L) heterosexual men. A positive PSE shift indicates a feminine bias, that is, a bias toward perceiving the PLWs as more feminine. Error bars: SEMs adjusted for individual differences; *: p<0.05; **: p≤0.01; ***: p≤0.005.

The online version of this article includes the following source data and figure supplement(s) for figure 4:

**Source data 1.** Experiments 3 and 5.

**Source data 2.** Experiment 4.

**Figure supplement 1.** Experiment 4 replicated the main findings of Experiment 3 that oxytocin modulates chemosensory decoding of femininity in high AQ heterosexual men in a dose-dependent manner.

or 24 IU vasopressin ($t_{23}$ = −0.24, p=0.81; *Figure 4E,F*). The results thus more directly pointed to the involvement of oxytocin, but not vasopressin, in the processing of chemosensory sexual information. Critically, they suggested that the effect of oxytocin was not monotonic. At 12 IU, intranasal oxytocin facilitated the decoding of the feminine information carried by estratetraenol in high AQ heterosexual men; but at 24 IU, the effect disappeared.

The nonmonotonic effect of oxytocin on chemosensory decoding of sex was striking, and we sought to replicate it before continuation. In Experiment 4, we adopted a fully double-blind placebo-controlled within-subject design and recruited another 24 high AQ (range: 25–39) heterosexual men, who underwent three drug conditions: saline, 12 IU oxytocin, and 24 IU oxytocin. Their results mirrored those of Experiment 3 (*Figure 4—figure supplement 1*). The combined data from Experiments 3 and 4 affirmed that high AQ heterosexual men were unsusceptible to estratetraenol at baseline (no drug treatment/saline; $t_{47}$ = 0.95, p=0.35; with no difference between no drug treatment and saline, p=0.95), yet showed a robust estratetraenol-induced gender perception bias when treated with 12 IU oxytocin ($t_{47}$ = 4.10, p<0.001, Cohen's d = 0.59) and not 24 IU oxytocin ($t_{47}$ = −0.025, p=0.98). There was overall a significant quadratic effect of intranasal oxytocin dose on estratetraenol-induced shift of gender judgment criterion in these participants ($F_{1, 47}$ = 6.58, p=0.014, partial $\eta^2$ = 0.12). We hence deduced that oxytocin level and the chemosensory processing of sexual cues follow an inverted-U-shaped relationship. It follows that low AQ individuals, who presumably have higher levels of endogenous oxytocin (*Koven and Max, 2014*; *Lancaster et al., 2015*; *Parker et al., 2014*), would benefit less, if any, from 12 IU oxytocin, and not from 24 IU oxytocin.

Furthermore, we predicted that their decoding of chemosensory sexual information would not be influenced by vasopressin regardless of the dose.

Our hypotheses were confirmed in Experiment 5, which was identical to Experiment 3 except that the participants were 24 heterosexual males with low AQ scores (range: 8–24). In sharp contrast to the high AQ men in Experiments 3 and 4, the low AQ individuals decoded the feminine information carried by estratetraenol in the absence of drug treatment ($t_{23}$ = 3.15, p=0.005, Cohen's d = 0.64; *Figure 4G,L*), again in line with the preliminary result from the low AQ individuals in Experiments 1 and 2 (*Figure 3—figure supplement 2*). The chemosensory effect was nonetheless diminished following the administration of 12 IU oxytocin ($t_{23}$ = 1.30, p=0.21; *Figure 4H,L*) and was abolished by the administration of 24 IU oxytocin ($t_{23}$ = 0.47, p=0.64; *Figure 4I,L*); that is, exogenous oxytocin at these doses hampered rather than facilitated the utilization of chemosensory feminine information in these low AQ individuals. Vasopressin, as expected, exerted no influence. The participants remained capable of detecting the feminine information carried by estratetraenol under the treatments of 12 IU and 24 IU vasopressin ($t_{23}$s = 4.02 and 3.10, ps = 0.001 and 0.005, Cohen's ds = 0.82 and 0.63, respectively; *Figure 4J–L*).

To further characterize the role of oxytocin in the chemosensory processing of sexual information, we analyzed the pooled PSEs across different combinations of oxytocin doses (no drug treatment/ saline, 12 IU oxytocin, 24 IU oxytocin) and olfactory stimuli (estratetraenol vs. carrier control) from both the high AQ and low AQ heterosexual men in Experiments 3–5. We found that relative to baseline (no drug treatment/saline), the high AQ and low AQ individuals showed opposite patterns when treated with 12 IU oxytocin, resulting in a significant three-way interaction among social proficiency, oxytocin treatment (no drug treatment/saline vs. 12 IU oxytocin) and olfactory condition ($F_{1, 70}$ = 5.65, p=0.020, partial $\eta^2$ = 0.075), on top of a strong main effect of olfactory condition ($F_{1, 70}$ = 20.63, p<0.0001, partial $\eta^2$ = 0.23) and a marginally significant interaction between social proficiency

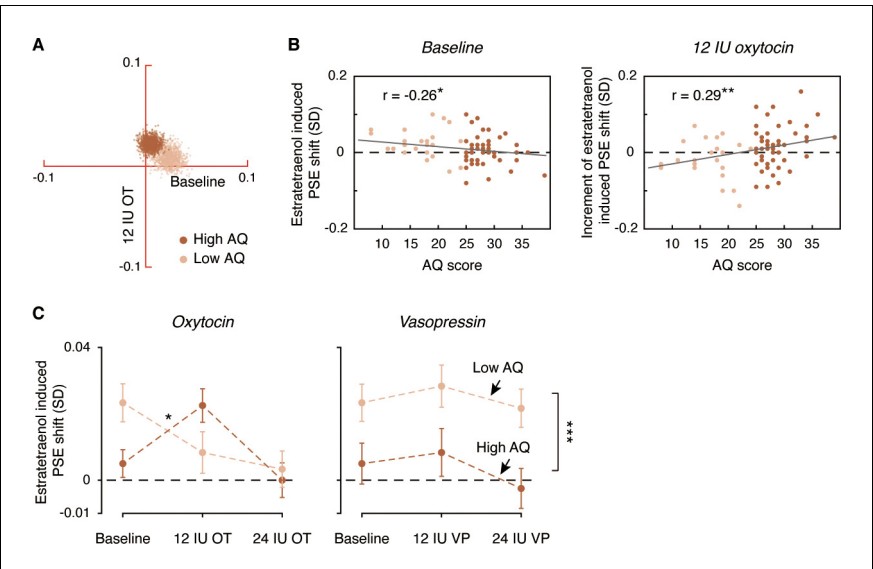

**Figure 5.** Comparison of dose-response relationships of oxytocin and vasopressin between high AQ and low AQ heterosexual men. (**A**) Bivariate distributions of bootstrapped sample means for high AQ (Experiments 3–4, 1000 dark brown dots) and low AQ (Experiment 5, 1000 light brown dots) heterosexual men plotted against the horizontal and vertical axes, representing estratetraenol-induced point of subjective equality (PSE) shifts at baseline (no drug treatment/saline) and following 12 IU intranasal oxytocin, respectively. (**B**) AQ score was negatively correlated with estratetraenol-induced PSE shift at baseline (left panel) and positively correlated with the increase of estratetraenol-induced PSE shift post 12 IU oxytocin treatment (right panel). (**C**) Overall, exogenous oxytocin modulated estratetraenol-induced PSE shift in manners that were dose-dependent and contingent upon the recipient's social proficiency (left panel), whereas exogenous vasopressin consistently showed no significant impact (right panel). A positive PSE shift indicates a feminine bias, that is, a bias toward perceiving the point-light walkers (PLWs) as more feminine. Error bars: SEMs adjusted for individual differences; *: p<0.05; **: p≤0.01; ***: p≤0.005.

and oxytocin treatment ($F_{1, 70}$ = 3.13, p=0.081). To illustrate these effects, we plotted in *Figure 5A* the central tendencies of the estratetraenol-induced PSE shifts (relative to the carrier control) at baseline (x axis) and under 12 IU oxytocin (y axis) from both the high AQ and low AQ heterosexual men in Experiments 3–5. The bootstrapped sample means of the two groups of men formed two discrete clusters: those of high AQ individuals (dark brown dots) fell around the vertical axis on the positive side, indicating that they utilized the feminine information carried by estratetraenol only after 12 IU oxytocin treatment, and those of low AQ individuals (light brown dots) fell around the horizontal axis on the positive side, indicating a significant effect of estratetraenol in them at baseline but not after 12 IU oxytocin treatment. A closer inspection of the data revealed that AQ score, while negatively correlated with estratetraenol-induced PSE shift at baseline ($r_{72}$ = −0.26, p=0.028), was positively correlated with the increase of estratetraenol-induced PSE shift post 12 IU oxytocin treatment (i.e. estratetraenol-induced PSE shift under 12 IU oxytocin minus that at baseline; $r_{72}$ = 0.29, p=0.014) (*Figure 5B*). In other words, one's social proficiency was predictive of his response to 12 IU oxytocin treatment. Under 24 IU oxytocin, both groups of men became insusceptible to estratetraenol (olfactory condition × social proficiency: $F_{1, 70}$ = 0.13, p=0.72; olfactory condition: $F_{1, 70}$ = 0.11, p=0.74; social proficiency: $F_{1, 70}$ = 0.072, p=0.79), consistent with the results of Experiment 1. Overall, as summarized in *Figure 5C*, the effect of exogenous oxytocin on the chemosensory processing of sexual cues was dose-dependent, nonmonotonic and contingent upon the recipient's social proficiency. By contrast, exogenous vasopressin did not exert any significant impact.

At the same time, the difference limens of the high AQ and low AQ heterosexual men in Experiments 3 and 5 did not differ ($F_{1, 46}$ = 0.078, p=0.78) and were comparable across all olfactory conditions and drug treatments (i.e. 12 IU oxytocin, 24 IU oxytocin, 12 IU vasopressin, 24 IU vasopressin, and no drug treatment) (olfactory condition × drug treatment: $F_{4, 184}$ = 1.48, p=0.21; olfactory condition: $F_{1, 46}$ = 1.00, p=0.32; drug treatment: $F_{4, 184}$ = 1.66, p=0.16). Their mood states, as reflected by self-reported ratings on the POMS, were also stable across olfactory conditions and drug treatments (total mood disturbance: ps = 0.76, 0.86, and 0.45, respectively; all subscales: ps > 0.05, corrected). We also specifically examined the difference limens and POMS ratings across different doses of oxytocin (no drug treatment/saline, 12 IU oxytocin and 24 IU oxytocin) and olfactory conditions for all the participants in Experiments 3–5, and obtained the same results (ps > 0.5). Moreover, these participants could not tell apart estratetraenol and the carrier solution alone by smell (mean accuracy = 0.36 vs. chance = 0.33, $t_{71}$ = 0.93, p=0.36). We therefore concluded that the chemosensory processing of sexual information took place below olfactory awareness. Through interactions with the oxytocin system, it shifted one's criterion, but not sensitivity, of gender perception, without significantly altering his transient mood state.

## Discussion

The current study represents an initial effort to unravel the neuroendocrine mechanisms underlying human chemo-signaling of sex. Data collected through formal psychophysical testing of 216 individuals over a total of 1056 testing sessions jointly demonstrate that the decoding of chemosensory sexual cues, including that of estratetraenol in heterosexual men and of androstadienone in homosexual men, is modulated by oxytocin instead of vasopressin in a dose-dependent manner, and is blocked by atosiban, a competitive antagonist of both oxytocin and vasopressin receptors (*Manning et al., 2012*; *Table 1*).

We are aware of a recent criticism of studies using androstadienone and/or estratetraenol that show a positive effect, which states that these studies were underpowered and problematic due to small sample sizes, lack of a priori evidence of effects, and lack of full replication, and the results are likely false positives (*Wyatt, 2015*). We, along with other researchers in the field (*Endevelt-Shapira et al., 2018*), do not agree with this criticism and believe that the current study presents a strong counter-argument. We had clear a priori evidence of the effects of androstadienone and estratetraenol from an earlier study (*Zhou et al., 2014*). Using the same task as in that study, we consistently replicated the original findings (*Table 1*), namely, androstadienone signals masculinity to homosexual men (Experiment 2) and estratetraenol signals femininity to heterosexual men (Experiments 1, 3, 4, and 5). The overall effect size was comparable to that of gender adaptation using visually presented faces or bodies (*Ghuman et al., 2010*). Moreover, whereas the natural occurring

**Table 1.** Summary of the effects of androstadienone and estratetraenol on the recipients' gender judgment criteria across various drug treatments.

Each cell represents results from 24 participants, respectively from Experiments 1 (heterosexual men), 2 (homosexual men), 3 (high AQ heterosexual men), and 5 (low AQ heterosexual men). −: no significant effect relative to the carrier control, +: a positive effect, that is, recipients biased toward perceiving the point-light walkers (PLWs) as more masculine (by androstadienone, A) or more feminine (by estratetraenol, E).

| | Baseline [*] | Atosiban | 24 IU OT | 24 IU VP |
|---|---|---|---|---|
| Heterosexual men | A− E+ | A− E− | A− E− | A− E+ |
| Homosexual men | A+ E− | A− E− | A− E− | A+ E− |

| | Baseline | 12 IU OT | 24 IU OT | 12 IU VP | 24 IU VP |
|---|---|---|---|---|---|
| High AQ heterosexual men | E−[†] | E+ [†] | E−[†] | E− | E− |
| Low AQ heterosexual men | E+ | E− | E− | E+ | E+ |

[*]Results from our earlier study (*Zhou et al., 2014*) for comparison.
[†]Results replicated in Experiment 4.

concentration of estratetraenol in sweat has not been measured, the concentration of androstadienone (500 µM) presented to the participants was similar to that in freshly produced apocrine sweat (mean = 0.44 nmol/µl = 0.44 × $10^{-3}$ mol/l = 440 µM) (*Gower et al., 1994*) and thus arguably ecologically relevant (although likely significantly higher than those encountered in non-intimate social interactions). It was indeed the reliability of these chemosensory effects that allowed us to probe into the underlying neuroendocrine mechanisms and uncover their modulation by oxytocin rather than vasopressin.

Oxytocin and vasopressin differ from each other at only two amino acid positions. Both are strongly implicated in a range of reproductive and social behaviors in animals as well as humans (*Carter, 1992*; *Carter, 2014*; *Donaldson and Young, 2008*). Many of the roles of oxytocin have been associated with female-typical behaviors in animals, and many of the behaviors associated with vasopressin have been demonstrated in males (*McCall and Singer, 2012*). Nonetheless, the specific behaviors they influence show extensive variation among different species, and to which extent the sex-related differences hold for humans is scantly known (*Donaldson and Young, 2008*). Here, we found in heterosexual and homosexual men that the utilization of the feminine information carried by estratetraenol, while 'male-typical', was not modulated by vasopressin regardless of its dose or the recipient's social proficiency, but was instead modulated by oxytocin, like the 'female-typical' utilization of the masculine information carried by androstadienone. These results hence argue against a clear-cut sex- or sexual orientation-based division of labor between the oxytocin and vasopressin systems in humans.

Importantly, we observed that the oxytocinergic modulation of human chemosensory decoding of sex was dose-dependent, non-monotonic, and roughly followed an inverted-U-shaped function, and that the dose effect of exogenous oxytocin depended on the recipient's social proficiency. At 12 IU, nasally administered oxytocin restored the ability to utilize the feminine information carried by estratetraenol in socially less proficient heterosexual men, yet hampered the very ability in socially proficient ones. Overall, the less socially proficient a man was, the less he utilized chemosensory sexual information at baseline, the more he became to do so after 12 IU oxytocin treatment, and vice versa. Increasing the dose of oxytocin to 24 IU eliminated the processing of the chemosensory sexual cues irrespective of the recipient's sexual orientation or social proficiency. It has been shown in rats that the dose-response curve for oxytocin is bell-shaped. Moderate doses of oxytocin facilitate, whereas high doses of oxytocin inhibit penile erections (*Argiolas and Gessa, 1991*). Similarly, moderate doses of oxytocin promote, whereas high doses of oxytocin disrupt social memory (*Benelli et al., 1995*). Moreover, a marked increase of oxytocin has been suspected to mediate or signal satiety and contribute to a refractory state in animals (*Carter, 1992*). In humans, there has also been hints that the effect of intranasal oxytocin is dose-dependent (*Cardoso et al., 2013*; *Quintana et al., 2017*). A recent study indicates that pretreatment blood oxytocin concentrations

predict treatment response to a set dosage of intranasal oxytocin in children with autism (*Parker et al., 2017*). Several others note that the influence of intranasal oxytocin on socially proficient individuals is complicated, and can be null or even antisocial (*Bartz et al., 2011*). Our findings dovetail with these reports. Moreover, they provide strong behavioral evidence for a non-monotonic effect of intranasal oxytocin that interacts with the recipient's social proficiency. While we tested healthy male volunteers, the non-monotonic property of oxytocin's effects has significant implications in the design of oxytocin-based treatment protocols for conditions like autism spectrum disorder.

Following the work of *Kosfeld et al., 2005*, the majority of studies on the effect of oxytocin in humans have used a single intranasal dose of 24 IU and shown a prosocial effect. Most have not simultaneously evaluated the effect of vasopressin or an antagonist of oxytocin or vasopressin receptors. In the limited studies noting that a higher dose of oxytocin produces a smaller effect, the phenomenon is speculated to be a result of cross-binding to vasopressin receptors that 'cancels out' the oxytocinergic effect (*Cardoso et al., 2013*). In our study, only at 12 IU did intranasal oxytocin significantly facilitate the decoding of sexual information in socially less proficient individuals. There was no significant effect of vasopressin irrespective of the dose. Thus, the lack of a positive effect of 24 IU oxytocin could not be explained by cross-binding of oxytocin to vasopressin receptors. Rather, we argue that it reflects the dose-response characteristics of oxytocin. Moreover, in view of the well-documented prosocial effect of 24 IU intranasal oxytocin, our results suggest that the optimal oxytocin level for the decoding of chemosensory sexual information differs from that for the promotion of prosocial behavior. After all, social behavior evolves as a result of effects upon the reproductive competition among group members (*Alexander, 1974*).

In rats, oxytocin has been found to enhance social recognition by modulating cortical control of early olfactory processing (*Oettl et al., 2016*). What neural pathways subserve the observed oxytocinergic modulation of human chemosensory communication? How does oxytocin act on sexually dimorphic sensory processing? These interesting questions await further research to clarify. Given the evolutionary and anatomical intimacy between the olfactory system and the neuroendocrine system, particularly through the hypothalamus (*Gorbman, 1995*), analyses of their interplays will open up new avenues for the regulations of both sensory perception of the external world and physiological processes within the human body.

## Materials and methods

### Participants

A total of 216 young male adults participated in the main study, 72 (mean age ± SD = 23.94 ± 1.37 years) in Experiment 1, 72 (22.15 ± 2.31 years) in Experiment 2, 24 (24.21 ± 1.87 years) in Experiment 3, 24 (23.38 ± 2.50 years) in Experiment 4, and 24 (24.00 ± 3.01 years) in Experiment 5. They provided ratings of their sexual orientation on the Kinsey scale (Kinsey Institute), where 0 is exclusively heterosexual, 3 is equally heterosexual and homosexual, and 6 is exclusively homosexual. They also completed the Autism Spectrum Quotient (AQ) (*Baron-Cohen et al., 2001*), a self-administered instrument that measures one's social proficiency, prior to the lab sessions. The participants in Experiments 1, 3, 4, and 5 had Kinsey scores $\leq$ 1 (97.2% had Kinsey scores = 0), whereas those in Experiment 2 had Kinsey scores $\geq$ 4 (5.26 ± 0.63). The participants in Experiments 3 and 4 had AQ scores $\geq$ 25 (28.37 ± 3.59 and 28.71 ± 2.87, respectively, out of 50 total), whereas those in Experiment 5 had AQ scores < 25 (17.00 ± 4.41). Women were not recruited due to pragmatic difficulties: The effect of oxytocin in women could be affected by menstrual phase (fluctuations of gonadal steroids) and the use of hormonal contraceptives (*Insel et al., 1993*; *Scheele et al., 2016*). Oxytocin also causes uterine contraction, which could be particularly problematic for women during early pregnancy. Sample sizes (n = 24 in each subgroup) were determined by G*Power to be adequate to detect a moderate effect of androstadienone or estratetraenol ($d \approx$ 0.6), at 80% power. The effect size was estimated based on an earlier study that employed almost identical stimuli and psychophysical testing procedures to those in the current study (*Zhou et al., 2014*). In essence, for each drug condition, we examined whether there was a significant effect of androstadienone or estratetraenol in a subgroup of 24 participants. All participants were healthy nonsmokers with normal or corrected-to-normal vision, normal sense of smell, and no respiratory allergy or upper respiratory infection at

the time of testing. They gave written informed consent to participate in procedures approved by the Institutional Review Board at Institute of Psychology, Chinese Academy of Sciences, and were unaware of the purposes of the experiments.

## Olfactory stimuli

The olfactory stimuli consisted of androstadienone (500 µM in 1% v/v clove oil propylene glycol solution, 5 ml), estratetraenol (500 µM in 1% v/v clove oil propylene glycol solution, 5 ml), and their carrier solution alone (1% v/v clove oil in propylene glycol, 5 ml) in Experiments 1 and 2, and the latter two, namely, estratetraenol and the carrier solution alone, in Experiments 3–5. The effectiveness of the clove oil carrier solution as a masker for the odors of androstadienone and estratetraenol was verified beforehand in an independent group of 48 healthy male nonsmokers in a standard triangular test (22.71 ± 2.37 years; 6 trials, mean accuracy ± SD = 0.33 ± 0.15 vs. chance = 0.33, p>0.99) (*Ye et al., 2019*; *Zhou et al., 2014*). They were presented in identical 40 ml polypropylene jars, each connected with two Teflon nosepieces via a Y-structure and coded by an individual not involved in the study. Participants were instructed to hold the jar with their non-dominant hand, position the nosepieces inside their nostrils, and continuously inhale through the nose and exhale through their mouth throughout each block of the experiments. Since the psychological effects of androstadienone is unrelated to one's sensitivity to its odor (*Lundström et al., 2003a*) and estratetraenol is generally regarded as odorless (*Lundström et al., 2003b*), we did not assess individuals' thresholds to androstadienone or estratetraenol without an odor mask.

## Visual stimuli

The visual stimuli were identical to those used in the aforementioned earlier study and were described in detail therein (*Zhou et al., 2014*). Briefly, parametric, gender-morphable PLWs (http://www.biomotionlab.ca/Demos/BMLwalker.html) (*Troje, 2002*) were generated with MATLAB and presented on a 22-inch LCD monitor using the psychophysics toolbox. Each walker (visual angle = 2.4°×7.8°) comprised 15 moving dots (0.2°×0.2°) depicting the trajectories of the major joints during walking. The gender was indexed by a normalized Z score on an axis that differentiated between actual male and female walkers in terms of a linear classifier. For each participant, the PLWs' gender varied in seven equal steps from 0.45 standard deviation (SD) into the female part of the axis to 0.45 SD into the male part of the axis, with the center being approximately perceived gender neutrality, which was individually set prior to the actual experiment in the absence of drug treatment and olfactory stimulus.

## Gender identification task and evaluation of mood states

We employed the same gender identification task as described in *Zhou et al., 2014*. In each trial (*Figure 1A*), participants viewed a 500 ms fixation cross (0.5°×0.5°) followed by a PLW presented for 500 ms (0.5 walking cycle) at a random location 0–1° away from fixation, and then pressed one of two buttons to indicate whether the walker was a male or a female. The next trial began immediately after a response was made. Each block consisted of 70 trials (7 PLWs × 10 repetitions in random order) and lasted about 3.5 min. The initial frame of each motion sequence was randomized.

Each participant completed multiple testing sessions, one session per day. On each day of testing, participants in Experiments 1 and 2 first completed 5 blocks of the gender identification task in the absence of olfactory stimulus (an empty jar was used instead) and drug treatment, which served as the baseline, and then 7 blocks 35 min after drug administration (see section below) while being continuously exposed to either androstadienone, estratetraenol, or their carrier solution alone, one on each day over 3 consecutive days in a counterbalanced manner. There was a break of at least 1 min in between every two blocks to eliminate fatigue and olfactory adaptation. The 7 experimental blocks as a whole typically took about 30 min to complete. Those in Experiments 3 and 5 were tested over 10 days. On each day, they similarly performed 5 baseline blocks in the absence of olfactory stimulus and drug treatment, and then 7 experimental blocks—either 35 min after drug administration (on 8 days) or 10 min after the completion of the baseline blocks without drug treatment (on 2 days, see section below)—under the continuous exposure to estratetraenol or the carrier solution alone, one on each day in a counterbalanced manner. Experiment 4 followed the same procedure as in Experiments 3 and 5 except that participants were tested over 6 days and received drug

treatment on each day. Following the experimental blocks, participants completed the Profile of Mood States (POMS) (*McNair et al., 1971*), a 65-item self-reported rating scale assessing transient, distinct mood states including tension, depression, fatigue, confusion, anger, and vigor, on each day of testing. The experimenter was not in the test room while the participants performed the tasks.

## Drug application

On each day of testing, participants in Experiments 1 and 2 self-administered a single intranasal dose of 24 IU of oxytocin, 24 IU of vasopressin, or 60 µg of atosiban (ProSpec, >99.0%, 98.0%, and 99.0% as determined by RP-HPLC, respectively, dissolved in saline; three puffs per nostril, each with 4 IU of oxytocin, 4 IU of vasopressin, or 10 µg of atosiban) after the completion of the baseline blocks of the gender identification task, in a between-subjects manner (*Figure 1D*). Atosiban is a desamino-oxytocin analogue and a competitive antagonist for both oxytocin and vasopressin receptors (*Manning et al., 2012*), and is close to oxytocin and vasopressin in both structure and molar mass (994.2, 1007.2, and 1084.2 g/mol, respectively). All three nonapeptides are plausibly centrally available when administered intranasally (*Born et al., 2002*; *Freeman et al., 2016*; *Liu et al., 2018*; *Lundin et al., 1986*). Since the IU for atosiban has not been established and 24 IU is the equivalent of about 48 µg oxytocin and 60 µg vasopressin, we chose to use 60 µg atosiban as a conservative dose to antagonize some of the central actions of oxytocin and vasopressin.

Those in Experiments 3 and 5 followed similar procedures, but each underwent five drug conditions over 10 days (within-subjects factor, *Figure 1E*), namely, 12 IU oxytocin (three puffs per nostril, each with 2 IU of oxytocin), 24 IU oxytocin, 12 IU vasopressin (three puffs per nostril, each with 2 IU of vasopressin), 24 IU vasopressin, and no drug treatment, where they received no nasal spray and were instructed to rest for 10 min before performing the experimental blocks of the gender identification task. Participants in Experiment 4 each underwent three drug conditions over 6 days (within-subjects factor), that is, saline, 12 IU oxytocin, and 24 IU oxytocin. For each participant, sessions with the same drug condition were held on consecutive days. The order of drug conditions was randomized across participants.

Fresh oxytocin, vasopressin, and/or atosiban solutions were made every 3 days during the period of data collection, such that for each participant in each experiment, the solution he received was prepared in less than 3 days before. The prepared solutions were stored in 10 ml sterilized nasal spray bottles at 4°C until usage and were coded by an individual not involved in the study.

## Statistical analyses

Responses from the gender identification task were first baseline normalized (mean shifting) per drug and olfactory condition for each participant to eliminate day-to-day variations in gender judgment criterion that were unrelated to the experimental manipulations. For each of the seven PLWs, the baseline adjusted proportion of 'male' responses $p'$ was calculated as $p' = p_{exp} - p_{base} + \bar{p}_{base}$, where $p_{exp}$ and $p_{base}$ are the averaged proportions of 'male' responses in the experimental blocks and the preceding baseline blocks on the same day, respectively, and $\bar{p}_{base}$ is the mean proportion of 'male' responses in the baseline blocks across the days of testing (3 days for Experiments 1 and 2, 10 days for Experiments 3 and 5, and 6 days for Experiment 4).

In Experiments 1 and 2, the baseline normalized gender judgments per olfactory condition per participant were subsequently fitted with a Boltzmann sigmoid function $f(x) = 1/(1 + \exp((x - x_0)/\omega))$ to better characterize the response criteria and sensitivities. $x_0$ corresponds to the PSE, at which the observer judged a PLW as male 50% of the time; half the interquartile range of the fitted function corresponds to difference limen, an index of his discrimination sensitivity. For each of the three drug treatment groups, we then performed paired sample t tests to compare the PSEs under the exposures to androstadienone and estratetraenol, respectively, with that under the exposure of the carrier solution alone. We also employed a standard bootstrapping procedure (*Davison and Hinkley, 1997*) to highlight the central tendencies of the androstadienone and estratetraenol induced PSE shifts (relative to carrier control) in heterosexual males (Experiment 1) and homosexual males (Experiment 2) across different drug conditions (*Figure 3*). Specifically, the original dataset of each subgroup (n = 24) of participants was randomly resampled with replacement to form a bootstrap sample of size 24. This procedure was repeated 1000 times, resulting in 1000 bootstrapped sample means per olfactory and drug condition per subgroup. In addition, repeated

measures ANOVAs were conducted on the PSEs, difference limens, and POMS ratings to compare the response criteria, discrimination sensitivities, and self-reported mood states between heterosexual and homosexual males (between-subjects factor) across olfactory conditions (within-subjects factor) and drug treatments (between-subjects factor).

Data from Experiments 3–5 were analyzed in similar manners. Unlike Experiments 1 and 2, there were two olfactory conditions (estratetraenol and carrier control) and five (Experiments 3 and 5: no drug treatment, 12 IU oxytocin, 24 IU oxytocin, 12 IU vasopressin, and 24 IU vasopressin) or three (Experiment 4: saline, 12 IU oxytocin, and 24 IU oxytocin) drug conditions. Both olfactory condition and drug condition were manipulated in a within-subjects fashion. We specifically compared estratetraenol-induced gender perception biases between socially less proficient (AQ scores $\geq$ 25, Experiments 3 and 4) and socially proficient heterosexual males (AQ scores < 25, Experiment 5) across drug conditions to characterize the dose-response properties of oxytocin and vasopressin in these two groups of participants.

All statistical tests were two-sided.

## Acknowledgements

We thank Yuli Wu for assistance. This work was supported by the National Natural Science Foundation of China (31830037 and 31422023), the Key Research Program of Frontier Sciences (QYZDB-SSW-SMC055) and the Strategic Priority Research Program (XDB32010200) of the Chinese Academy of Sciences, and Beijing Municipal Science and Technology Commission.

## Additional information

### Funding

| Funder | Grant reference number | Author |
|---|---|---|
| National Natural Science Foundation of China | 31830037 | Wen Zhou |
| National Natural Science Foundation of China | 31422023 | Wen Zhou |
| Chinese Academy of Sciences | QYZDB-SSW-SMC055 | Wen Zhou |
| Chinese Academy of Sciences | XDB32010200 | Wen Zhou |
| Beijing Municipal Science and Technology Commission | | Wen Zhou |

The funders had no role in study design, data collection and interpretation, or the decision to submit the work for publication.

### Author contributions

Kepu Chen, Data curation, Formal analysis, Validation, Investigation, Writing - original draft; Yuting Ye, Data curation, Formal analysis, Validation, Investigation, Writing - review and editing; Nikolaus F Troje, Software, Writing - review and editing; Wen Zhou, Conceptualization, Supervision, Funding acquisition, Methodology, Writing - original draft, Writing - review and editing

### Author ORCIDs

Yuting Ye (iD) https://orcid.org/0000-0001-9068-8045
Wen Zhou (iD) https://orcid.org/0000-0001-6730-2116

### Ethics

Human subjects: Written informed consent and consent to publish were obtained from all participants in accordance with ethical standards of the Declaration of Helsinki (1964). The study was approved by the Institutional Review Board at Institute of Psychology, Chinese Academy of Sciences (H18029).

Decision letter and Author response
Decision letter https://doi.org/10.7554/eLife.59376.sa1
Author response https://doi.org/10.7554/eLife.59376.sa2

## Additional files

### Supplementary files

- Transparent reporting form

### Data availability

All data analyzed during this study are included in the manuscript and supporting files. Source data files have been provided for Figures 2, 3 and 4.

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
