## [Decision Letter]

**Acceptance summary:**

We are so pleased that you have made a necessarily complex paradigm involving several areas of expertise understandable to the general reader in service of an exciting advance of our understanding of chemosensory communication in humans. Clearly we are not moths. Yet even though human chemo-communication is not dramatic it nonetheless exists and influences perception, decisions, behavior in nuanced ways such as the compelling example that you demonstrate here.

**Decision letter after peer review:**

Thank you for submitting your article "Oxytocin mediates human chemosensory communication of sex in a dose-dependent manner" for consideration by *eLife*. Your article has been reviewed by four peer reviewers including Peggy Mason as the Reviewing Editor and Reviewer #1, and the evaluation has been overseen Catherine Dulac as the Senior Editor. The following individual involved in review of your submission has agreed to reveal their identity: Gün R. Semin (Reviewer #2).

The reviewers have discussed the reviews with one another and the Reviewing Editor has drafted this decision to help you prepare a revised submission.

All reviewers found the work to be interesting, fundamental and innovative. Hormones (OXT, VAS), gender perception using point light walking, and chemosensation intersect in this study making it inherently complex and somewhat challenging to the reader. The first point of revision is to clarify and hand-hold a bit more through the results and also to explain some of the expectations which were not obvious to the reviewers and are unlikely to be obvious to the general reader.

Beyond clarification, please also respond to the specific suggestions in the full reviews attached below.

Reviewer #1:

This innovative study examines the role of oxt and vas in gender perception using point light walking and administered female and male smells. The study is complex and difficult to keep track of but that is inherent to the design and the authors do a good job. I found the data compelling and the interpretations interesting.

In Figure 2E,J and Figure 3, the graphs should show some metric of the data rather than the shift so that the no drug condition and its variability can be illustrated.

Subsection “Oxytocin, vasopressin and subconscious chemosensory decoding of sex in heterosexual and homosexual men” “it follows that individuals with different endogenous oxytocin level would differ in their susceptibility to such chemosignals as well as to the effect of exogenous oxytocin.” Why does this follow?

Reviewer #2:

Overall, this is a very impressive study that logically builds across a set of 5 experiments a convincing case to uncover the neuroendocrine mechanisms involved in the chemosensory communication of sex and show that oxytocin and vasopressin have different roles to play in this process.

Some trivial observations:

1) I would have appreciated information about which the criteria they used -if any – to decide about sample sizes.

2) It would have been useful if the authors would have argued as to why they did not use heterosexual and lesbian women participants?

Reviewer #3:

In this research, the authors investigated the biological substrate(s) of chemosensory decoding of femininity, specifically focusing on the oxytocin and, closely related, vasopressin systems based on their roles in reproductive and social behavior. Results showed that oxytocin, but not vasopressin, plays a causal role in chemosensory communication in humans. The paper was well-written and I particularly liked how they used an oxytocin agonist AND antagonist to support their claims, as well as vasopressin to speak to discriminant validity. This work makes an important contribution to our understanding of the biological mechanisms that support human social information processing.

My first concern has to do with sample size, which seems on the small side. That said, I do appreciate how difficult it is to conduct these kinds of drug administration studies, and the fact that the authors ran multiple studies and reported consistent effects across studies, so I am torn. Perhaps the authors could provide additional information on statistical power. The authors do address statistical power (subsection “Participants”):

"…Sample sizes (n = 24 in each subgroup) were determined by G*Power to be adequate to detect a moderate effect of androstadienone or estratetraenol (d ≈ 0.6), at 80% power. The effect size was estimated based on an earlier study that employed almost identical stimuli and psychophysical testing procedures to those in the current study (Zhou et al., 2014)."

However, it seems that they are only reporting the power to detect the effect of androstadienone and estratetraenol on chemosensory communication, NOT the moderation by drug, or the moderation by individual differences (AQ, sexual orientation). Can the authors please speak to these issues? Especially for Study 3 where there appear to be 10 conditions (12 IU OT, 24 IU OT, 12 IU AVP, 24 IU AVP, PL x estratetraenol vs. carrier).

My second concern has to do with the author's assertion that less socially proficient individuals have lower levels of endogenous and the reason why oxytocin should be helpful to them is BECAUSE they have lower levels of oxytocin. For example, as the authors write:

Abstract: "…and contingent upon the recipients' social proficiency – a partial manifestation of their endogenous oxytocin level."

Discussion section: "…such that the dose effect of exogenous oxytocin depended on the recipient's social proficiency, which in turn partially reflected his endogenous oxytocin level (Parker et al., 2014)."

Discussion section: "Moreover, they provide strong behavioral evidence for a non-monotonic effect of intranasal oxytocin that interacts with the recipient's social proficiency or endogenous oxytocin level."

This last statement is particularly problematic given that the authors did not actually measure participants endogenous OT, but the statement makes it seem like they did.

While there is evidence linking endogenous OT levels with social proficiency, to my knowledge, no one has demonstrated that exogenous OT selectively benefits less socially proficient individuals because it alters endogenous OT levels. Given this, the authors might want to re-think their rationale for Experiments 3 and 4. Actually, there is plenty of empirical evidence for the selectively beneficial effects of oxytocin for individuals who are less socially proficient (e.g., Luminet et al., 2011; Radke and de Bruijn; 2015; Feeser et al., 2015), specifically as assessed by the AQ (Bartz et al., 2010; 2019)-I think citing that research is sufficient justification to look at AQ as a moderator in Experiment 3 and 4. Of course, the authors can mention the endogenous OT-ASD findings; I just wouldn't make explicit claims about mechanism as it seems like they are (unnecessarily) going out on a limb.

Reviewer #4:

This study aims to test whether intranasal administration of the neuropeptides Oxytocin (OXT) and Vasopressin (AVP) can mediate the effect of two putative human chemosignals – androsta-4,16,-dien-33-one (AND) and estra-1,3,5(10),16-tetraen-3-ol (EST). To test this hypothesis, the authors used a behavioral task named PLW, in which participants determine the gender (female or male) of a dot-figure. In the manuscript, they detailed 5 experiments which provide evidence for a link between OXT and the effect of EST.

In general, the manuscript is novel, and provides an important contribution, yet there are a few points which should be addressed:

1) The manuscript is a bit hard to follow. Though it is divided to sets of experiments per test focus, it is hard to follow the line of thought which lead to each experiment, what was the hypothesis raised and the way they wanted to test it.

2) In the main Experiments (1-5) the statistical method used is paired t-test. Data of this sort should be statistically tested using one statistical test (e.g. ANOVA instead of multiple t-tests), with factors of odor and participant.

3) Were there corrections for multiple comparisons applied? Please mention this in the text.

4) It is not clear whether all experiments were double-blind. Were both-experimenter and subject not aware of the drugs and odors administrated in all experiments? If not, this should be clearly acknowledged.

5) According to what did the authors choose a cut-off of 25 score for AQ? There is no supporting for this cut-off in the reference they provided, and in a previous study by the same group (2018) they chose a cut-off of 20. This should be clarified.

6) In regards to the olfactory stimuli, in the Materials and methods section it is mentioned "Participants were instructed to hold the jar with their non-dominant hand, position the nosepieces inside their nostrils and continuously inhale through the nose and exhale through their mouth throughout each block of the experiments". Did the authors verify using breathing measurements (e.g. spirometer) whether the participants actually inhaled from their nose and exhaled from their mouth? If not, this could add variability to the results and should be mentioned.

7) In the Results section it is mentioned as "The three olfactory stimuli were perceptually indiscriminable as assessed in a separate panel of 48 male participants (mean triangular discrimination accuracy = 0.33 vs. chance = 0.33, t47 = 0.02, p > 0.9)". However, in the methods section, it is mentioned "The effectiveness of the clove oil carrier solution as a masker for the odors of androstadienone and estratetraenol was verified beforehand in an independent group of 48 healthy male nonsmokers (22.71 {plus minus} 2.37 yrs; triangular test, mean accuracy {plus minus} SD = 0.33 {plus minus} 0.15 vs. chance = 0.33, p > 0.99)….…Since the psychological effects of androstadienone is unrelated to one's sensitivity to its odor (Lundström et al., 2003) and estratetraenol is generally regarded as odorless (Lundstrom, Hummel and Olsson, 2003), we did not assess individuals' sensitivities to androstadienone or estratetraenol alone". It is not clear whether the authors did actually test, or how did they test, if participants could discriminate or perceive the smell of these odorants, and was this tested separately per each odor stimulus? There is evidence that AND has an odor [Keller et al., 2007] and can be perceived as unpleasant to participants. If there were perceptual differences this may affect the interpretation of the results.

8) Throughout the manuscript the authors refer to "social proficiency" and participants' basal OXT levels, this without measuring their basal OXT levels. This issue has further implications due to their claim that OXT affects the response to EST, however, this claim becomes circular when basal levels are not measured nor taken into account when reaching conclusions of the effect of OXT.

9) Please rephrase the sentence where you suggest that using homosexual men is a replacement for women. (Discussion section)

[Editors' note: further revisions were suggested prior to acceptance, as described below.]

Thank you for resubmitting your article "Oxytocin mediates human chemosensory communication of sex in a dose-dependent manner" for consideration by *eLife*. Your revised article has been reviewed by three peer reviewers including Peggy Mason as the Reviewing Editor and Reviewer #1, and the evaluation has been overseen by Catherine Dulac as the Senior Editor.

This manuscript has been responsive to the reviews. However, the writing remains dense and unfortunately errors in referring to the figures do not help the reader to decode this revision. Please take this opportunity to clean up the text and figures so that the revision can be fully appreciated and evaluated.

Reviewer #1:

This study remains quite complex and unfortunately errors in referring to the figures do not help the reader to decode this revision.

I would like the errors to be fixed so that a clear read can be had. Critical errors are in the referring to Figure 2, which while only one of 5 figures, is foundational to the story. Confusing the reader during the establishment of the basic paradigm is problematic.

First, it would appear that the men and women labels at the bottom of Figure 2F are reversed. If they are not, then I am confused. I looked at this easily 5-10 times and with those labels the figure does not compute.

Results section:

Figure 2B should be 2F?

should be Figure 2B G E J

Should be Figure 2D,I not G I

Why focus on VAS instead of OXT?

Reviewer #3:

Overall I think the authors did a fine job responding to the reviewers' comments, questions and suggestions. I just have a few remaining items:

1) Clarity about participants and experiments:

I raised this issue before-and I appreciate the changes the authors made-but I still feel they need to be more transparent about participants/studies/experiments and sessions:

"Psychophysical data collected from 216 heterosexual and homosexual men over 1,056 sessions consistently showed that such chemosensory communications of sex were blocked by a competitive antagonist of both oxytocin and vasopressin receptors called atosiban, administered nasally"

This still isn't entirely transparent as the sample size collapses across 5 experiments, thus giving the impression of greater statistical power than is warranted (per experiment/test). Please present sample size for each individual experiment here, or at least indicate that the 216 ps comprise 5 experiments.

"Systematic comparisons of these psychometric curves across 5 experiments totaling 1,056 sessions enabled…"

As per above, please state number of ps here for transparency.

Also, I find the term “sessions” to be confusing. Maybe this is a convention specific to their research area, but, to me, “sessions” suggests testing sessions, whereas it seems like the authors are referring to trials on the task?

I think reviewer 4 made a similar comment, but I also found the overall design to be unclear:

Are Experiments 1 and 2 identical expect that Experiment 1 has heterosexual men and Experiment 2 has homosexual men as participants? If so, why are they called two different experiments since the results are discussed side by side and, in fact, the authors combine the data anyway with the omnibus ANOVA?

More generally, is this one “Study” with multiple “Experiments” embedded within the overall Study? The Participants section is written this way (i.e., "A total of 216 young male adults participated in the main study…") but the way the Experiments are presented in the main text I had the impression that they were separate studies, with different participants and different designs, not one overall study, with experiment referring to testing different effects.

2) Predictions about oxytocin:

As detailed below, I was a bit surprised by some of the findings and I think the authors could do a better job walking readers through. Note: I am in no way suggesting that the authors should state hypotheses that they did not have a priori, but I think they need to take more care laying the groundwork for readers. See my suggestions below:

"…could be affected in either direction by the administration of oxytocin or vasopressin."

I was surprised that the authors did not have specific predictions for OT, given that they did have predictions about atosiban. Since OT is an agonist, wouldn't one expect the opposite effect of the antagonist atosiban? Again, I am not advocating that the authors make predictions post hoc, but I think they need to clarify *why* they did not have predictions for OT since it's not obvious (i.e., given they had predictions for atosiban).

"Smelling estratetraenol and androstadienone, relative to the carrier solution alone, failed to respectively influence the gender perception criteria (indexed by the PSEs) of the atosiban-treated heterosexual and homosexual men (t23s = 1.37 and 0.88, ps = 0.18 and 0.39, respectively; Figure 2C, D, I, J). This appeared to be the case for those treated with 24 IU oxytocin as well (t23s = 0.17 and -1.29, ps = 0.87 and 0.21, respectively; Figure 2E, F, I, J)."

I was also surprised that oxytocin and atosiban produced the same effect, one being an agonist and the other being an antagonist. I appreciate that the authors address this in the discussion, and I basically find their rationale to be compelling, but my sense is that they gloss over this surprising finding here and elsewhere (see below). I think they could do a better job of guiding the reader along by NOT glossing over this point. E.g., something like "Interestingly, this appeared to be the case for those treated with 24 IU oxytocin as well (t23s = 0.17 and -1.29, ps = 0.87 and 0.21, respectively; Figure 2E, F, I, J)."

I have similar issues with the following statements:

"…were subserved by similar neuroendocrine mechanisms that were disrupted by intranasal atosiban, the competitive antagonist of both oxytocin and vasopressin receptors, and unaffected by intranasal vasopressin."

Didn't oxytocin exert the same effect as atosiban? But oxytocin is not mentioned.

"The question remained as to why the administration of 24 IU oxytocin exempted the participants from the influences of the chemosignals, and we explored it in more detail."

I would suggest the following modification for transparency: "the question remained as to why the administration of 24IU oxytocin exerted the same effect as atosiban…"

Rather than glossing over this surprising finding, I think it would be more effective to acknowledge it head on.

---

## [Author Response]

All reviewers found the work to be interesting, fundamental and innovative. Hormones (OXT, VAS), gender perception using point light walking, and chemosensation intersect in this study making it inherently complex and somewhat challenging to the reader. The first point of revision is to clarify and hand-hold a bit more through the results and also to explain some of the expectations which were not obvious to the reviewers and are unlikely to be obvious to the general reader.

We have clarified the rationales, hypotheses, and results of the experiments in the revised manuscript. Please also refer to our responses to reviewer 1’s points 1 and 2, reviewer 3’s points 2 and 8, and reviewer 4’s points 1-3 below.

Reviewer #1:This innovative study examines the role of oxt and vas in gender perception using point light walking and administered female and male smells. The study is complex and difficult to keep track of but that is inherent to the design and the authors do a good job. I found the data compelling and the interpretations interesting.

We thank the reviewer for the positive assessment of our work.

1) In Figure 2E,J and Figure 3, the graphs should show some metric of the data rather than the shift so that the no drug condition and its variability can be illustrated.

The PSE shifts in Figure 2E,J and Figure 3 are relative to their respective control conditions (clove oil carrier solution alone). We have now included a new figure in the revised manuscript (Figure 3—figure supplement 1) that illustrates the central tendencies of the PSEs under each combination of olfactory (androstadienone, estratetraenol, carrier control) and drug conditions (60 μg of atosiban, 24 IU oxytocin, 24 IU vasopressin). We note that the PSEs under the exposure to the carrier solution alone were comparable across drug treatments and between heterosexual and homosexual men (drug treatment: F_2, 138_ = 0.092, p = 0.91; sexual orientation: F_1, 138_ = 0.011, p = 0.92; interaction: F_2, 138_ = 0.22, p = 0.80).

2) Subsection “Oxytocin, vasopressin and subconscious chemosensory decoding of sex in heterosexual and homosexual men” “it follows that individuals with different endogenous oxytocin level would differ in their susceptibility to such chemosignals as well as to the effect of exogenous oxytocin.” Why does this follow?

We have changed the sentence to “If oxytocin plays a role in the processing of estratetraenol (in heterosexual males) and androstadienone (in homosexual males), given the heterogenous effects of oxytocin on individuals with different levels of social proficiency (Bartzvet al., 2011), specifically as assessed by the AQ (Bartz et al., 2019; Bartz et al., 2010), it follows that individuals with different AQ scores could differ in their susceptibility to such chemosignals as well as to the effect of exogenous oxytocin” in subsection “Oxytocin, vasopressin and subconscious chemosensory decoding of sex in heterosexual and homosexual men” of the revised manuscript. Please also refer to our response below to a related point raised by reviewer 3 (reviewer 3’s point 2).

Reviewer #2:Overall, this is a very impressive study that logically builds across a set of 5 experiments a convincing case to uncover the neuroendocrine mechanisms involved in the chemosensory communication of sex and show that oxytocin and vasopressin have different roles to play in this process.

We appreciate the reviewer’s positive remarks.

Some trivial observations:1) I would have appreciated information about which the criteria they used -if any – to decide about sample sizes.

In essence, for each drug condition, we examined whether there was a significant effect of androstadienone or estratetraenol in a subgroup of 24 participants. Sample sizes (n = 24 in each subgroup) were determined by G*Power to be adequate to detect a moderate effect of androstadienone or estratetraenol (d ≈ 0.6), at 80% power. The effect size was estimated based on an earlier study that employed almost identical stimuli and psychophysical testing procedures to those in the current study (Zhou et al., 2014). This is clarified in the Materials and methods section of the revised manuscript. See also our response below to a related point raised by reviewer 3 (reviewer 3’s point 1).

2) It would have been useful if the authors would have argued as to why they did not use heterosexual and lesbian women participants?

The effect of oxytocin in women could be affected by menstrual phase (fluctuations of gonadal steroids) and the use of hormonal contraceptives (Insel et al., 1993; Scheele et al., 2016). Oxytocin also causes uterine contraction, which could be particularly problematic for women during early pregnancy. It is thus pragmatically difficult to examine the effect of oxytocin in women. These are now clarified in the Materials and methods section of the revised manuscript.

Reviewer #3:In this research, the authors investigated the biological substrate(s) of chemosensory decoding of femininity, specifically focusing on the oxytocin and, closely related, vasopressin systems based on their roles in reproductive and social behavior. Results showed that oxytocin, but not vasopressin, plays a causal role in chemosensory communication in humans. The paper was well-written and I particularly liked how they used an oxytocin agonist AND antagonist to support their claims, as well as vasopressin to speak to discriminant validity. This work makes an important contribution to our understanding of the biological mechanisms that support human social information processing.

We thank the reviewer for the positive assessment of our work and the constructive suggestions.

1) My first concern has to do with sample size, which seems on the small side. That said, I do appreciate how difficult it is to conduct these kinds of drug administration studies, and the fact that the authors ran multiple studies and reported consistent effects across studies, so I am torn. Perhaps the authors could provide additional information on statistical power. The authors do address statistical power (subsection “Participants”):"…Sample sizes (n = 24 in each subgroup) were determined by G*Power to be adequate to detect a moderate effect of androstadienone or estratetraenol (d ≈ 0.6), at 80% power. The effect size was estimated based on an earlier study that employed almost identical stimuli and psychophysical testing procedures to those in the current study (Zhou et al., 2014)."However, it seems that they are only reporting the power to detect the effect of androstadienone and estratetraenol on chemosensory communication, NOT the moderation by drug, or the moderation by individual differences (AQ, sexual orientation). Can the authors please speak to these issues? Especially for Study 3 where there appear to be 10 conditions (12 IU OT, 24 IU OT, 12 IU AVP, 24 IU AVP, PL x estratetraenol vs. carrier).

As we were unaware of any previous study that had examined the modulation of chemosensory decoding of sex by oxytocin/vasopressin or individual differences, it was not possible to estimate a priori the effect sizes of their interactions. Our study was therefore powered to detect the effect of androstadienone or estratetraenol in each subgroup of participants (n = 24) under different drug treatments. Put differently, for each drug condition, we examined whether there was a significant effect of androstadienone or estratetraenol in a subgroup of 24 participants. This is now clarified in the Materials and methods section of the revised manuscript.

In Experiments 1 and 2, drug treatment served as a between-subjects factor, our analyses of the interactions among olfactory condition, drug treatment, and sexual orientation (Figure 2 and Figure 3) were based on the data from a total of 144 participants and 432 testing sessions. In Experiments 3-5, drug treatment served as a within-subjects factor, our analyses of the interactions among olfactory condition, drug treatment, and social proficiency (Figure 4 and Figure 5) were based on the data from a total of 72 participants and 624 testing sessions. We would like to note that our sample sizes are comparable to, if not larger than, those in some earlier studies that addressed the interplays between drug and individual differences (e.g., (Bartz et al., 2019; Bartz et al., 2010; Feeser et al., 2015; Scheele et al., 2014; Spengler et al., 2017)).

2) My second concern has to do with the author's assertion that less socially proficient individuals have lower levels of endogenous and the reason why oxytocin should be helpful to them is BECAUSE they have lower levels of oxytocin. For example, as the authors write:Abstract: "…and contingent upon the recipients' social proficiency – a partial manifestation of their endogenous oxytocin level."Discussion section: "…such that the dose effect of exogenous oxytocin depended on the recipient's social proficiency, which in turn partially reflected his endogenous oxytocin level (Parker et al., 2014)."Discussion section: "Moreover, they provide strong behavioral evidence for a non-monotonic effect of intranasal oxytocin that interacts with the recipient's social proficiency or endogenous oxytocin level."This last statement is particularly problematic given that the authors did not actually measure participants endogenous OT, but the statement makes it seem like they did.While there is evidence linking endogenous OT levels with social proficiency, to my knowledge, no one has demonstrated that exogenous OT selectively benefits less socially proficient individuals because it alters endogenous OT levels. Given this, the authors might want to re-think their rationale for Experiments 3 and 4. Actually, there is plenty of empirical evidence for the selectively beneficial effects of oxytocin for individuals who are less socially proficient (e.g., Luminet et al., 2011; Radke and de Bruijn; 2015; Feeser et al., 2015), specifically as assessed by the AQ (Bartz et al., 2010; 2019)-I think citing that research is sufficient justification to look at AQ as a moderator in Experiments 3 and 4. Of course, the authors can mention the endogenous OT-ASD findings; I just wouldn't make explicit claims about mechanism as it seems like they are (unnecessarily) going out on a limb.

We fully agree and would like to thank the reviewer for directing us to these studies. We have made changes to the text in the Abstract and in the Discussion section of the revised manuscript accordingly to be more stringent with our expressions. Some of the mentioned studies are also cited in the revised manuscript.

Reviewer #4:This study aims to test whether intranasal administration of the neuropeptides Oxytocin (OXT) and Vasopressin (AVP) can mediate the effect of two putative human chemosignals – androsta-4,16,-dien-33-one (AND) and estra-1,3,5(10),16-tetraen-3-ol (EST). To test this hypothesis, the authors used a behavioral task named PLW, in which participants determine the gender (female or male) of a dot-figure. In the manuscript, they detailed 5 experiments which provide evidence for a link between OXT and the effect of EST.In general, the manuscript is novel, and provides an important contribution, yet there are a few points which should be addressed:

We thank the reviewer for the overall positive evaluation of our work and the detailed comments and suggestions.

Essential revisions:1) The manuscript is a bit hard to follow. Though it is divided to sets of experiments per test focus, it is hard to follow the line of thought which lead to each experiment, what was the hypothesis raised and the way they wanted to test it.

The rationales and hypotheses for Experiments 1-2, 3, 4, and 5 are now respectively stated in the Results section of the revised manuscript. Please also refer to our responses above to two related points raised by reviewer 3 (points 2 and 8).

2) In the main experiments (1-5) the statistical method used is paired t-test. Data of this sort should be statistically tested using one statistical test (e.g. ANOVA instead of multiple t-tests), with factors of odor and participant.

The results of ANOVAs are now reported in subsection “Oxytocin, vasopressin and subconscious chemosensory decoding of sex in heterosexual and homosexual men” (Experiments 1 and 2) and in subsection “Dose-dependent modulation of chemosensory communication of sex by oxytocin but not vasopressin in high and low AQ individuals” (Experiments 3-5) of the revised manuscript.

3) Were there corrections for multiple comparisons applied? Please mention this in the text.

As mentioned in our response to reviewer 3’s point 1, our study was designed to examine whether there was a significant effect of androstadienone or estratetraenol in each subgroup of participants under each drug condition. The comparisons between androstadienone/estratetraenol and carrier control were planned (see also our response to this reviewer’s point 1) and hence did not require corrections for multiple comparisons. Moreover, the reported effects in each experiment would remain statistically significant if we were to apply Bonferroni correction.

4) It is not clear whether all experiments were double-blind. Were both-experimenter and subject not aware of the drugs and odors administrated in all experiments? If not, this should be clearly acknowledged.

As mentioned in the Materials and methods section of the original manuscript, both the odor solutions and the drug solutions were coded by an individual not involved in the study. All experiments were double-blind. In the “no drug” condition in Experiments 3 and 5, where both the experimenter and the participants knew no drug was administered, they were still blind to the identities of the olfactory stimuli.

5) According to what did the authors choose a cut-off of 25 score for AQ? There is no supporting for this cut-off in the reference they provided, and in a previous study by the same group (eLife 2018) they chose a cut-off of 20. This should be clarified.

At the group level, 24 IU oxytocin exempted the heterosexual men in Experiment 1 from the influence of estratetraenol and the homosexual men in Experiment 2 from the influence of androstadienone. Considering that oxytocin exerts a more pronounced and typically positive effect on socially less proficient individuals as measured by the AQ (Bartz et al., 2019; Bartz et al., 2010; Bartz et al., 2011), we adopted a relatively stringent criterion for high AQ individuals — AQ scores ≥ 25, i.e. 1 SD or more above the reported mean for males (Baron-Cohen et al., 2001) — in our supplementary analysis of the data from Experiments 1 and 2 and in Experiments 3 and 4 in hopes to better capture the effects of social proficiency and oxytocin in chemosensory communication of sex. This is now clarified in subsection “Oxytocin, vasopressin and subconscious chemosensory decoding of sex in heterosexual and homosexual men” of the revised manuscript.

We have also re-examined the data from Experiments 3-5 using 20 as the AQ cut-off and obtained similar patterns of results as those reported in the manuscript.

6) In regards to the olfactory stimuli, in the Materials and methods section it is mentioned "Participants were instructed to hold the jar with their non-dominant hand, position the nosepieces inside their nostrils and continuously inhale through the nose and exhale through their mouth throughout each block of the experiments". Did the authors verify using breathing measurements (e.g. spirometer) whether the participants actually inhaled from their nose and exhaled from their mouth? If not, this could add variability to the results and should be mentioned.

We verified that the participants inhaled through the nose and exhaled through the mouth by monitoring the surface of the liquid in the odor jar via a camera. As mentioned in our response to reviewer 3’s point 7, the order of olfactory conditions was counterbalanced across participants and that of drug conditions randomized across participants. The procedure controlled the effects of nuisance variables.

7) In the Results section it is mentioned as "The three olfactory stimuli were perceptually indiscriminable as assessed in a separate panel of 48 male participants (mean triangular discrimination accuracy = 0.33 vs. chance = 0.33, t47 = 0.02, p > 0.9)". However, in the methods section, it is mentioned "The effectiveness of the clove oil carrier solution as a masker for the odors of androstadienone and estratetraenol was verified beforehand in an independent group of 48 healthy male nonsmokers (22.71 {plus minus} 2.37 yrs; triangular test, mean accuracy {plus minus} SD = 0.33 {plus minus} 0.15 vs. chance = 0.33, p > 0.99)….…Since the psychological effects of androstadienone is unrelated to one's sensitivity to its odor (Lundström et al., 2003) and estratetraenol is generally regarded as odorless (Lundstrom, Hummel and Olsson, 2003), we did not assess individuals' sensitivities to androstadienone or estratetraenol alone". It is not clear whether the authors did actually test, or how did they test, if participants could discriminate or perceive the smell of these odorants, and was this tested separately per each odor stimulus? There is evidence that AND has an odor [Keller et al., 2007] and can be perceived as unpleasant to participants. If there were perceptual differences this may affect the interpretation of the results.

We verified the effectiveness of clove oil carrier solution (1% v/v clove oil propylene glycol solution) as a masker for the odors of androstadienone (500 μM) and estratetraenol (500 μM) in an independent group of 48 healthy male nonsmokers. Specifically, we tested whether they could discriminate among androstadienone in clove oil carrier solution, estratetraenol in clove oil carrier solution, and clove oil carrier solution alone, using a standard triangular test (6 trials per participant). Detailed testing procedures have been described elsewhere (Ye et al., 2019; Zhou et al., 2014). Briefly, in each trial, blindfolded participants were presented with three smells, two identical (comparison) and the other one different (target), and reported which one was the odd one out. Each smell served as the comparison in 1/3 of the trials and the target in 1/3 of the trials. Results showed that the three olfactory stimuli were perceptually indiscriminable, mean accuracy = 0.33 vs. chance = 0.33, p > 0.9, consistent with earlier reports (Ye et al., 2019; Zhou et al., 2014). Put differently, the odor of androstadienone or estratetraenol (if any) was effectively masked by clove oil. We did not assess individuals’ thresholds to androstadienone or estratetraenol without an odor mask.

These are now clarified in the Materials and methods section of the revised manuscript.

8) Throughout the manuscript the authors refer to "social proficiency" and participants' basal OXT levels, this without measuring their basal OXT levels. This issue has further implications due to their claim that OXT affects the response to EST, however, this claim becomes circular when basal levels are not measured nor taken into account when reaching conclusions of the effect of OXT.

This point is related to reviewer 3’s point 2. Please refer to our response to that point above. We would also like to note that several studies have found a link between social proficiency and endogenous oxytocin level (Koven and Max, 2014; Lancaster et al., 2015; Parker et al., 2014).

9) Please rephrase the sentence where you suggest that using homosexual men is a replacement for women. Discussion section

We have removed this sentence in the revised manuscript.

[Editors' note: further revisions were suggested prior to acceptance, as described below.]

Reviewer #1:This study remains quite complex and unfortunately errors in referring to the figures do not help the reader to decode this revision.I would like the errors to be fixed so that a clear read can be had. Critical errors are in the referring to Figure 2, which while only one of 5 figures, is foundational to the story. Confusing the reader during the establishment of the basic paradigm is problematic.1) First, it would appear that the men and women labels at the bottom of Figure 2F are reversed. If they are not, then I am confused. I looked at this easily 5-10 times and with those labels the figure does not compute.

The x-axis in each of Figure 2A-D and F-I, Figure 4A-E and G-K, and Figure 4—figure supplement 1A-C represents PLW’s physical gender. As mentioned in the Introduction and the Materials and methods section of the original main text, PLW’s gender was indexed by a normalized Z score on an axis that differentiated between actual male and female walkers in terms of a linear classifier, and ranged from -0.45 (female-like) to 0.45 (male-like). Overall, the more masculine a PLW was, the more likely it was judged as “male” (larger value on the y-axis that represents the proportion of “male” responses). We have clearly labeled the x-axis in these figures in the revised manuscript.

We realize that the PLW’s physical gender and the male and female symbols, appearing in the original Figure 2E and J, Figure 3, Figure 3—figure supplement 1, Figure 4F and L, Figure 4—figure supplement 1D, and Figure 5A, are confusing. The male and female symbols were meant to represent a masculine bias (a bias towards perceiving the PLWs as more masculine, i.e., a negative PSE shift) and a feminine bias (i.e., a positive PSE shift) in participants’ gender perception, respectively. We have since removed the male and female symbols from all figures. In addition, we have clarified and illustrated the relationships among PLW’s physical gender, proportion of “male” responses, PSE, and PSE shift in the Results section of the revised main text and in the revised Figure 1B and C

2) Results section:Figure 2B should be 2F? Should be Figure 2B,G,E,J; Should be Figure 2D,I not G,I

We apologize for the errors. We tried different ways to organize and present the results of Experiments 1 and 2 and in this process mistakenly referred to the wrong subfigures in the text. Per reviewer 3’s suggestions (reviewer 3’s points 1d and 2), we have reorganized this part of the Results section and have made sure that the correct figures are referred to in the text.

3 Why focus on VAS instead of OXT?

As mentioned in the original manuscript, we set out to explore if high AQ and low AQ participants in Experiments 1 and 2 differed in their susceptibility to the chemosignals. Since no significant effect of androstadienone or estratetraenol was observed under the 24 IU oxytocin condition or 60 µg atosiban condition, we focused on the 24 IU vasopressin condition where the effects of androstadienone and estratetraenol were comparable to those obtained earlier without drug treatment (Zhou et al., 2014). This is now clarified in the Results section of the revised main text.

Reviewer #3:Overall I think the authors did a fine job responding to the reviewers' comments, questions and suggestions. I just have a few remaining items:1) Clarity about participants and experiments:I raised this issue before-and I appreciate the changes the authors made-but I still feel they need to be more transparent about participants/studies/experiments and sessions:1a) "Psychophysical data collected from 216 heterosexual and homosexual men over 1,056 sessions consistently showed that such chemosensory communications of sex were blocked by a competitive antagonist of both oxytocin and vasopressin receptors called atosiban, administered nasally"This still isn't entirely transparent as the sample size collapses across 5 experiments, thus giving the impression of greater statistical power than is warranted (per experiment/test). Please present sample size for each individual experiment here, or at least indicate that the 216 ps comprise 5 experiments.

The Abstract has a word limit of 150 words. We have stated in the revised Abstract that psychophysical data were collected from 216 heterosexual and homosexual men across 5 experiments.

1b) "Systematic comparisons of these psychometric curves across 5 experiments totaling 1,056 sessions enabled…"As per above, please state number of ps here for transparency

We have included the sample size of each experiment in the Introduction of the revised main text.

1c) Also, I find the term “sessions” to be confusing. Maybe this is a convention specific to their research area, but, to me, “sessions” suggests testing sessions, whereas it seems like the authors are referring to trials on the task?

“Sessions” does mean testing sessions. We have made it explicit in the revised manuscript. Specifically, as mentioned in the original manuscript, each participant was tested on multiple days (3 days in Experiments 1 and 2, 10 days in Experiments 3 and 5, 6 days in Experiment 4) and completed 1 testing session per day. Each session consisted of 12 blocks (5 baseline blocks and 7 experimental blocks) and each block consisted of 70 trials. The results reported in our manuscript are based on data collected from a total of 1,056 testing sessions or 1,056 person-days, which amount to 887,040 trials.

1d) I think reviewer 4 made a similar comment, but I also found the overall design to be unclear:1d-i) Are Experiments 1 and 2 identical expect that Experiment 1 has heterosexual men and Experiment 2 has homosexual men as participants? If so, why are they called two different experiments since the results are discussed side by side and, in fact, the authors combine the data anyway with the omnibus ANOVA?

Experiment 2 was conducted after we analyzed the results of Experiment 1, which presented a complex picture. Antagonizing oxytocin and vasopressin receptors with atosiban abolished the known effect of estratetraenol on heterosexual males (i.e. biasing them towards perceiving the PLWs as more feminine), yet 24 IU oxytocin appeared to produce the same effect. On the flip side, the administration of 24 IU vasopressin did not significantly alter the processing of estratetraenol in heterosexual males –– they remained biased towards perceiving the PLWs as more feminine under the exposure of estratetraenol, to the same extent as when no drug was administered. We wondered if this pattern of drug influences would hold for the chemosensory decoding of masculine information carried by androstadienone. To this end, we turned to homosexual males in Experiment 2, who had been shown to be subconsciously biased by androstadienone, but not estratetraenol, in making gender judgments (Zhou et al., 2014). Experiment 2 was identical to Experiment 1 except for the participants’ sexual orientation. These are now clarified I the Results section of the revised main text.

We presented the results of Experiments 1 and 2 side by side in the original manuscript in hopes to make the section more concise and also to make it easier to compare the results of heterosexual and homosexual men. We have realized that this only made the rationales and results of the experiments hard to unpack. We have rewritten this part of the Results section to clarify the rationale and results of each experiment. With regard to the use of omnibus ANOVAs on the pooled data of two or more experiments, please refer to our response below to this reviewer’s point 1d-ii.

1d-ii) More generally, is this one “Study” with multiple “Experiments” embedded within the overall Study? The Participants section is written this way (i.e., "A total of 216 young male adults participated in the main study…") but the way the Experiments are presented in the main text I had the impression that they were separate studies, with different participants and different designs, not one overall study, with experiment referring to testing different effects.

The distinction between “study” and “experiment” can be subtle. We chose to present our work as 1 study with 5 experiments because, as mentioned in the original manuscript, we had one single goal that was to examine the roles of oxytocin and vasopressin in chemosensory communications of sex through androstadienone and estratetraenol. Except for Experiment 1, the design and interpretation of each experiment was guided by the findings of the previous experiment(s). The combined results of all 5 experiments, encapsulated by the omnibus ANOVAs on the pooled data of Experiments 1 and 2 and of Experiments 3-5, jointly demonstrated that the decoding of chemosensory sexual cues is modulated by oxytocin instead of vasopressin in a dose-dependent manner.

2) Predictions about oxytocin:As detailed below, I was a bit surprised by some of the findings and I think the authors could do a better job walking readers through. Note: I am in no way suggesting that the authors should state hypotheses that they did not have a priori, but I think they need to take more care laying the groundwork for readers. See my suggestions below:2a) "…could be affected in either direction by the administration of oxytocin or vasopressin."I was surprised that the authors did not have specific predictions for OT, given that they did have predictions about atosiban. Since OT is an agonist, wouldn't one expect the opposite effect of the antagonist atosiban? Again, I am not advocating that the authors make predictions post hoc, but I think they need to clarify *why* they did not have predictions for OT since it's not obvious (i.e., given they had predictions for atosiban).

We had no specific prediction for oxytocin or vasopressin as we did not know whether one or both of them play a role in the processing of chemosensory sexual cues. It was also difficult to predict the directions of their effects (if any), as both positive and negative effects of oxytocin and vasopressin have been reported in the literature, depending on dose, context, and personal characteristics (Bartz et al., 2011; Carter, 2014; Donaldson and Young, 2008). We predicted that the administration of atosiban would block the effects of the chemosignals if such effects were regulated by the oxytocin/vasopressin system, because atosiban is a competitive antagonist of both oxytocin and vasopressin receptors. These are now clarified in the Results section of the revised main text.

2b) "Smelling estratetraenol and androstadienone, relative to the carrier solution alone, failed to respectively influence the gender perception criteria (indexed by the PSEs) of the atosiban-treated heterosexual and homosexual men (t23s = 1.37 and 0.88, ps = 0.18 and 0.39, respectively; Figure 2C, D, I, J). This appeared to be the case for those treated with 24 IU oxytocin as well (t23s = 0.17 and -1.29, ps = 0.87 and 0.21, respectively; Figure 2E, F, I, J)."I was also surprised that oxytocin and atosiban produced the same effect, one being an agonist and the other being an antagonist. I appreciate that the authors address this in the discussion, and I basically find their rationale to be compelling, but my sense is that they gloss over this surprising finding here and elsewhere (see below). I think they could do a better job of guiding the reader along by NOT glossing over this point. E.g., something like "Interestingly, this appeared to be the case for those treated with 24 IU oxytocin as well (t23s = 0.17 and -1.29, ps = 0.87 and 0.21, respectively; Figure 2E, F, I, J)."

Suggestion well taken. We have done so in the Results section of the revised main text. See also our response to this reviewer’s point 1d-i.

2c) I have similar issues with the following statements:"…were subserved by similar neuroendocrine mechanisms that were disrupted by intranasal atosiban, the competitive antagonist of both oxytocin and vasopressin receptors, and unaffected by intranasal vasopressin."Didn't oxytocin exert the same effect as atosiban? But oxytocin is not mentioned."The question remained as to why the administration of 24 IU oxytocin exempted the participants from the influences of the chemosignals, and we explored it in more detail."I would suggests the following modification for transparency: "the question remained as to why the administration of 24IU oxytocin exerted the same effect as atosiban…"Rather than glossing over this surprising finding, I think it would be more effective to acknowledge it head on.

We have revised these sentences accordingly in subsection “Oxytocin, vasopressin and subconscious chemosensory decoding of sex in heterosexual and homosexual men” of the revised main text. The sentences now read as follows:

“… were subserved by similar neuroendocrine mechanisms that were disrupted by intranasal atosiban –– the competitive antagonist of both oxytocin and vasopressin receptors, as well as by 24 IU oxytocin, and were unaffected by 24 IU vasopressin. Since vasopressin did not seem to play a role (participants’ response patterns to the chemosignals under 24 IU vasopressin were comparable to those previously obtained without drug treatment), by deduction, such mechanisms involved oxytocin. The question remained as to why the administration of 24 IU oxytocin, like atosiban, exempted the participants from the influences of the chemosignals, and we explored it in more detail.”

Please also refer to our responses to this reviewer’s points 1d-i, 2a and 2b.